



# A remote sensing-based dataset to characterize the ecosystem functioning and functional diversity of a Biosphere Reserve: Sierra Nevada (SE Spain)

5  Beatriz P. Cazorla[1,2], Javier Cabello[1,2], Andrés Reyes[1], Emilio Guirado[1,3], Julio Peñas[1,4], Antonio J. Pérez-Luque[5,6], Domingo Alcaraz-Segura[1,4,5]

[1]Andalusian Center for the Assessment and Monitoring of Global Change, University of Almería, 04120, Almería, Spain

[2] Department of Biology and Geology, University of Almería, 04120, Almería, Spain

[3]Andalusian Research Institute in Data Science and Computational Intelligence, University of Granada, 18071, Granada, Spain.

[4] Department of Botany, University of Granada, Av. de Fuentenueva, s/n 18071, Granada, Spain

[5] iecolab. Andalusian Institute for Earth System Research (IISTA-CEAMA) – University of Granada, Avda. Mediterráneo s/n, E-18006, Granada, Spain.

[6] Terrestrial Ecology Research Group, Department of Ecology, Faculty of Science, University of Granada, Av. Fuentenueva s/n, Granada, E-18071 Spain

*Correspondence to*: Beatriz P. Cazorla (b.cazorla@ual.es) and Domingo Alcaraz-Segura (dalcaraz@ugr.es)

20  **Abstract**

Conservation Biology faces the challenge of safeguarding the ecological processes that sustain biodiversity. Characterization and evaluation of these processes can be carried out through attributes or functional traits related to the exchanges of matter and energy between vegetation and the atmosphere. Nowadays, the use of satellite imagery provides useful methods to produce a spatially continuous characterization of ecosystem functioning and processes at regional scales. Our dataset characterizes the patterns of ecosystem functioning in Sierra Nevada (Spain) from the vegetation greenness dynamics captured through the spectral vegetation index EVI (Enhanced Vegetation Index) since 2001 to 2018 (product MOD13Q1.006 from MODIS sensor). First, we provided three Ecosystem Functional Attributes (EFAs) (i.e., descriptors of annual primary production, seasonality, and phenology of carbon gains), as well as their integration into a synthetic mapping of Ecosystem Functional Types (EFTs). Second, we provided two measures of functional diversity: EFT richness and EFT rarity. Finally, in addition to the yearly maps, we calculated interannual summaries, i.e., means and inter-annual variabilities. Examples of research and management applications of these data sets are also included to highlight the value of EFAs and EFTs to improve the understanding and monitoring ecosystem processes across environmental gradients. The datasets are available in two open-source sites (PANGAEA: https://doi.pangaea.de/10.1594/PANGAEA.904575 (Cazorla et al. 2019) and http://obsnev.es/apps/efts_SN.html), and bring to scientists, managers and the general public valuable information on the first characterization of the functional diversity at ecosystem level developed in a Mediterranean hotspot. Sierra Nevada represents an exceptional ecology laboratory of field conditions, where a long-term monitoring (LTER) program was established 10 years ago. The data availability on biodiversity, climate, ecosystem services, hydrology, land-use changes and management practices from Sierra Nevada, will allow to explore the relationships between these other environmental data and ecosystem functional data that we provide in this work.

## 1 Introduction

45  A better characterization of the functional dimension of biodiversity is required to develop management approaches that ensure nature contributions to human well-being (Jax, 2010). To achieve this goal, it is necessary to have a set of essential variables that characterize and monitor ecosystem functioning (Pereira et al., 2013). Such variables are basic to understand the dynamics of ecological systems (Petchey and Gaston, 2006), the links



between biological diversity and ecosystem services (Balvanera et al., 2006; Haines-Young and Potschin, 2010),
and the mechanisms of ecological resilience (Mouchet et al., 2010). In addition, the use of ecosystem functioning
variables has been demanded to assess functional diversity at large scales with the aim of measuring the Biosphere
integrity (Mace et al., 2014; Steffen et al., 2015), one of the most challenging planetary boundaries to quantify
(Steffen et al., 2015). Despite the importance of ecosystem functioning variables, and the conceptual frameworks
developed to promote their use (Pettorelli et al., 2018), they have seldom been incorporated to ecosystem
monitoring in protected areas (but see Alcaraz-Segura et al., 2009; Fernández et al., 2010; Cabello et al., 2016).

Characterization and evaluation of ecosystem functioning can be carried out through attributes or functional traits
related to the exchanges of matter and energy between vegetation and the atmosphere (Mueller-Dombois and
Ellenberg, 1974). Nowadays, the use of satellite imagery provides useful methods to produce a spatially
continuous characterization of ecosystem functioning and functional diversity at local (Fernández et al., 2010),
regional (Alcaraz-Segura et al., 2006, 2013) or global scales (Ivits et al., 2013). Theoretical and empirical models
support the relationship between spectral indices derived from satellite images (*e.g.* Enhanced Vegetation Index,
EVI) and essential functional variables of ecosystems, such as primary production, evapotranspiration, surface
temperature, or albedo (Running et al., 2000; Pettorelli et al., 2005; Fernández et al., 2010; Lee et al., 2013).
Among them, primary production is considered the most integrative and essential indicator of ecosystem
functioning (Virginia and Wall, 2001; Pereira et al., 2013), since it has an important role in the carbon cycle (i.e.,
it is the energy input to the trophic web and therefore, the driving force behind many ecological processes).
Moreover, primary production offers a comprehensive response to environmental changes, and constitutes a
synthetic indicator of ecosystem health (Costanza et al., 1992; Skidmore et al., 2015).

To characterize ecosystem functioning through spectral vegetation indices, we can use the approach based on
Ecosystem Functional Types (EFTs), defined as patches of the land surface that share similar dynamics in the
exchanges of matter and energy between the biota and the physical environment (Paruelo et al., 2001; Alcaraz-
Segura et al., 2006). EFTs are derived from three Ecosystem Functional Attributes (EFAs) that describe the
seasonal dynamics of carbon gains: annual mean (a surrogate of annual primary production, the most essential
and integrative indicator of ecosystem functioning), annual standard deviation (a descriptor of seasonality or the
differences between the growing and non-growing seasons), and the annual date of maximum (a phenological
indicator of when in the year is the growing period centered). Since the concept appeared in 2001 (Paruelo et al.,
2001), the EFT  approach (or equivalent approaches) has exponentially grown   to characterize functional
heterogeneity from local to global scales (Alcaraz-Segura et al., 2006; Karlsen et al., 2006; Duro et al., 2007;
Fernández et al., 2010; Geerken 2009; Alcaraz-Segura et al., 2013; Ivits et al., 2013; Cabello et al., 2013; Pérez-
Hoyos et al., 2014; Müller et al., 2014; Wang and Huang, 2015; Villarreal et al., 2018; Coops et al., 2018; Mucina,
2019).

This article aims to illustrate how EFAs and EFTs can be used to assess the spatio-temporal heterogeneity and
inter-annual variability of ecosystem functioning in protected areas based on the vegetation dynamics captured
through spectral vegetation indices (e.g. EVI). We introduce as a proof of concept the case of Sierra Nevada
Biosphere reserve (SE Spain) from 2001 to 2018. First, for each year, we provide three Ecosystem Functional
Attributes (EFAs) (i.e., annual primary production, seasonality and phenology of carbon gains), as well as their
integration into a synthetic mapping of Ecosystem Functional Types (EFTs). Second, we present two measures of
functional diversity: EFT richness and EFT rarity. Finally, in addition to the yearly maps, we calculated
interannual summaries, i.e., inter-annual means and inter-annual variability, to show the average conditions as
well as the most stable and variable zones along the period (workflow in Fig. 2).

## 2 Methods

### 2.1 Site Description

Sierra Nevada (Andalusia, SE Spain) is a mountainous region covering more than 2,000 km$^2$ with an elevation
range of between 860 and 3,482 m a.s.l (Fig. 1). It is considered one of the most important biodiversity hotspots





in the Mediterranean region (Blanca et al., 1998; Cañadas et al., 2014), hosting 105 endemic plant species for a total of 2,353 taxa of vascular plants (33% and 20% of Spanish and European flora, respectively; Lorite 2016). Forest cover in Sierra Nevada is dominated by pine plantations (*Pinus halepensis* Mill., *Pinus pinaster* Ait., *Pinus nigra* Arnold subsp. *salzmannii* (Dunal) Franco, and *Pinus sylvestris* L.) covering approximately 40,000 ha. Most of them were planted in the period 1960–1980. The main native forests of Sierra Nevada are dominated by the
evergreen holm oak *Quercus ilex* subsp. *ballota* (Desf.) Samp. occupying low and medium mountain areas (8,800 ha.), and by the deciduous Pyrenean oak *Quercus pyrenaica* Willd ranging from 1,100 to 2,000 m a.s.l. (about 2,000 ha). Autochthonous pine *Pinus sylvestris* L. var. *nevadensis* H. Christ forests can also be found in small patches with low tree cover in the treeline. Above the treeline, plant communities of the Oromediterranean and Crioromediterranean belts (above 1,800-2,000 m.) are dominated by chamaephytes and hemicryptophytes
(scrublands, grasslands, and cliff and scree communities), being the habitat to many endemic species. Sierra Nevada receives legal protection and international recognition in multiple ways: MAB Biosphere Reserve (1986), Natural Park (1989),National Park (1999), Important Bird Area (2003), Special Area of Conservation (Natura 2000 network, 2012), and it is included in the IUCN Green List of Areas (2014) and in the Spanish LTER network (Zamora et al. 2017). The main economic activities in this mountain region are agriculture,
tourism, livestock raising, beekeeping, mining, and skiing (Bonet et al., 2010).

In Sierra Nevada, vegetation studies have mainly been developed considering a compositional perspective (phytosociological method) or successional perspective (vegetation series). These studies have been very useful for describing the vegetation heterogeneity at mesoscale (Loidi, 2017), for characterizing habitats of conservation
importance (EU Directive 92/43/EEC), and for developing forest restoration policies (Valle et al., 2003). However, these approaches are difficult to monitor the effects of environmental changes and management actions, to understand the environmental gradients at protected area scale that drive biodiversity patterns, and to evaluate the role of ecosystems as suppliers of benefits to society (Cabello et al., 2019).

**2.2 Satellite images of Vegetation Indices (MOD13Q1 data product)**

The characterization of ecosystem functioning in Sierra Nevada was based on the temporal dynamics of the Enhanced Vegetation Index (EVI) from 2001 to 2018. Specifically, we used the MOD13Q1.006 product of the MODIS sensor (Moderate Resolution Imaging Spectroradiometer) on board NASA's Terra satellite (Didan 2015). This product provides maximum value composite images every 16 days (23 images per year) at 231 meters spatial resolution and are downloadable from NASA's LP DAAC (Land Processes Distributed Active Archive Center)
(http://lpdaac.usgs.gov/lpdaac/get_data) and in Google Earth Engine (DOI: https://doi.org/10.5067/MODIS/MOD13Q1.006 ). Values of EVI*10,000 are given as real numbers between 0 and 10,000.

**2.3 Calculating Ecosystem Functional Attributes (EFAs)**

We identified three EFAs that are known to capture most of the variance in the time series of vegetation indices and that are biologically meaningful (Paruelo et al., 2001; Alcaraz-Segura et al., 2006, 2009). These attributes were calculated from the EVI seasonal curve or annual dynamics. From the EVI seasonal curve of each year, we identified three functional attributes: the EVI annual mean (EVI_mean; an estimator of primary production), the EVI seasonal Standard Deviation (EVI_sSD; a descriptor of seasonality, i.e., the differences between the growing
and non-growing seasons), and the date of maximum EVI (EVI_DMAX; a phenological indicator of the month with maximum EVI) (Fig.3). To summarize the EFAs of the 2001-2018 period, we calculated the inter-annual mean and the inter-annual variability for each attribute.

**2.4 Identifying Ecosystem Functional Types (EFTs)**

EFTs were identified by synthesizing in a single map the variability contained in the three EFAs following a similar approach to Alcaraz-Segura et al. (2013). The range of values of each EFA was divided into four intervals, giving a potential number of 64 EFTs ($4 \times 4 \times 4$). For EVI_DMAX, the four intervals agreed with the four seasons of the year. For EVI_mean and EVI_sSD, we extracted the first, second, and third quartiles for each year and then





calculated the inter-annual mean of each quartile (means of the 18-year period) (Table 1). These fixed limits between EFT classes were applied to each year. To summarize the EFTs of the 2001–2018 period, we calculated the most frequent EFT of the period (i.e., the EFT mode for each pixel). To name EFTs, we used two letters and a number: the first capital letter indicates net primary production (EVI_mean), increasing from A to D; the second small letter represents seasonality (EVI_SD), decreasing from a to d; the numbers are a phenological indicator of the growing season (EVI_DMAX), with values 1-spring, 2-summer, 3-autumn, 4-winter.


**2.5 Characterizing Ecosystem Functional Diversity**

To characterize ecosystem functional diversity, we used EFT richness and EFT rarity. EFT richness was calculated for each year by counting the number of different EFTs in a 4×4-pixel moving window (924 x 924 m) around each pixel (top-left center pixel of the 4x4 Kernel) (modified from Alcaraz-Segura et al., 2013). Then, the average

richness map of all years was obtained. EFT rarity was calculated for each year as the relative extension of each EFT compared to the most abundant EFT (Equation 1) (Cabello et al., 2013). Then, the average rarity map of all years was obtained.

$$Rarity\ of\ EFT_i = (Area\_EFTmax – Area\_EFT_i)/Area\_EFTmax\ (Equation\ 1)$$


where Area_EFTmax is the area occupied by the most abundant EFT and Area_EFT$i$ is the area of the $i$ EFT being evaluated, with $i$ ranging from 1 to 64.

**2.6 Stability in ecosystem functioning**

To identify the most stable and variable areas (either due to inter-annual fluctuations or to directional trends) in ecosystem functioning, we provide three approaches. First, we calculated the inter-annual variability of each EFA (coefficient of variation for EVI_mean and EVI_sSD, and circular standard deviation for EVI_DMAX). Second, we recorded the number of different EFTs that occurred in the same pixel in the period 2001-2018. Third, to

consider the changes not only at the pixel but also at the landscape level, the Jaccard similarity index (Jaccard, 1901) (Equation 2) was used in 4×4-pixel moving windows (924 x 924 m).

$$Jaccard\ Index = (the\ number\ in\ both\ sets) / (the\ number\ in\ either\ set) * 100$$

The same formula in notation is (Equation 3):
$$J(X,Y) = |X \cap Y| / |X \cup Y|$$
In Steps:
1) Count the number of EFTs which are shared between both windows; 2) Count the total number of EFTs in both windows (shared and unshared); 3) Divide the number of shared EFTs 1) by the total number of EFTs 2); 4)

Multiply the number found in 3) by 100.

This measure represents how similar is the EFT composition that occurs in each window throughout the entire time series (2001-2018). For each window, the Jaccard index was calculated among all possible combinations of years and then the interannual average of all calculated indices was obtained. Dissimilarity was calculated as

(Equation 4):

$$Dissimilarity = 1 - Jaccard\ Index$$

Dissimilarity values range from 0 to 1, with 1 being the highest degree of dissimilarity in composition and relative

abundance of EFTs and 0 being absent.



## 3 Results and Discussion

### 3.1 Available dataset

Overall, the collection of datasets provides a characterization of ecosystem functioning and ecosystem functional diversity in Sierra Nevada Biosphere Reserve (SE Spain) through remote sensing. To broaden the use of data, first, we provide data in .tif format. Second, we have incorporated rendered versions of all layers as required by Google Earth Pro (called "filename..._forGoogleEarthVisualization.tif") for visualization. And third, we have also developed an ad-hoc visualization platform for all the layers.

All data are available yearly (2001-2018) and summarized for the period, in EPSG:4326 WGS84.

The dataset is structured in three main subsets of variables: Ecosystem Functional Attributes, Ecosystem Functional Types, and Ecosystem Functional Diversity (see Table 2). For each variable there are two groups of data (two subfolders): one containing the yearly variables, and another one containing the summaries for the 18-year period.

### 3.2 Ecosystem Functional Attributes patterns

Functional attributes of productivity, seasonality and phenology showed a clear altitudinal pattern. Productivity (EVI_mean) was much lower in the Crioro- and Oromediterranean bioclimatic belts than in the Supra- and Mesomediterranean belts. Productivity also decreased from west to east (Fig. 4a). Seasonality (EVI_sSD) was high in the Supramediterranean, decreasing in Meso-, and Thermomediterranean belts, and in Crioro- and Oromediterranean (Fig. 4b). Phenology (EVI_DMAX) was characterized by a dominant summer peak in vegetation greenness in the Crioro- and Oromediterranean belts, and a late spring peak in the Supra- and Mesomediterranean belts. Dry and semi-arid thermomediterranean areas of the south and east showed greeness peaks in early autumn and winter months (Fig. 4c).

### 3.3 Ecosystem Functional Type patterns

As a result of the combination of the three functional attributes of the canopy, productivity, seasonality and phenology, represented in Fig. 4 a-c, we obtained the EFTs map (Fig. 4d) that includes a synthetic characterization of the spatial patterns of ecosystem functioning. A total of 64 classes were observed. The most abundant EFT presented the maximum greenness in spring, with productivity values from low to intermediate and with all possible combinations of seasonality: Aa1, Ba1, Cb1, Cd1, Ba1, and Cc1 accumulated 30% of the surface. On the contrary, the rarest EFTs were Ba4, Aa4 characterized by medium or low productivity, high seasonality and maximum greenness in winter.

Crioro and oromediterranean areas presented EFTs with low and intermediate productivity, high seasonality and moments of maximum greenness mainly in summer, but also in spring. Here, extreme conditions characterized by scarce soil (Peinado et al., 2019), high solar radiation, extreme temperatures, winds, snow and ice, give rise to a short vegetative period. This results in scarce vegetation cover, controlled by low temperatures, which can only occur in summer, being the plant growth time, hence these areas have been referred to as "cold desert" (Blanca et al., 2019). The supra- and mesomediterranean levels had associated EFTs of intermediate-high productivity, medium-low seasonality and maximum green moment in spring and autumn (e.g., Cc1-3) (Fig. 4d). The supramediterranean is characterized by the presence of deciduous species, e.g., oak groves associated with the most productive and seasonal ecosystem functional type of the study area, with maximum in spring (EFT Da1). In the dry and semi-arid thermomediterranean of the eastern end, characterized by thermophilic species, which hardly suffer from frost, a different functional behaviour of the ecosystems was detected. The functioning of this area showed low values of productivity, medium-low seasonality and maximum greenness of the vegetation in spring or winter (e.g., Ac1-4). Here, the main control of ecosystem functioning is water availability, presenting plant species with a fast response to scarce water inputs (Cabello et al., 2012).





### 3.4 Stability in ecosystem functioning


The interannual variability ranged from 1 to 17 different EFTs over the 18-year period in the same pixel (Fig. 5a). The number of EFTs observed in the same pixel over 18 years was higher in the supra- and mesomediterranean levels, coinciding with the altitudinal range where interannual climate variability is most affected (e.g., they may present a lot of snow in cold years and be affected by drought in dry and warm years). The eastern end of the semi-arid thermomediterranean also highlighted with high inter-annual variability, where there exists a greater climate fluctuation and where small changes in precipitation produce large changes in the dynamics of primary


production (Houérou et al., 1988; Cabello et al., 2012), as well as the area burned in 2005 near Lanjarón, where the fire eliminated the vegetation that has been regenerating since then. On the other hand, the most stable vegetation types interannual, i.e., those that changed the least during the period, were located in the meso-oromediterranean and crioromediterranean levels, specifically, the oak and borreguil vegetation types, ecosystems that are not subject to anthropic presence (e.g., low forest management and low presence of livestock).


The results of the inverse of the Jaccard coefficient to obtain the dissimilarity or functional changes between years in the composition of EFTs over the 2001-2018 period (Fig. 5b), showed an altitudinal pattern where the dissimilarity between EFTs was lower in the oro and cryoromediterranean levels, as well as in the mesomediterranean oak groves (functional stability already shown by other authors, i. e. Requena-Mullor et al,


2018). This pattern of dissimilarity increased towards lower levels, finding the highest values of dissimilarity (or greater change) in areas where changes in land use and management are major (Zamora et al., 2016), such as autochthonous pine forests on dolomites, coniferous repopulations and meso- and thermomediterranean holm oaks. In addition, the eastern end of the Sierra Nevada had an area with low dissimilarity values, that is, where there were not many changes over the years and when they occurred they were towards very similar EFTs.


### 3.5 Functional diversity at ecosystem level

Richness oscillated between 1 and 13 EFTs. Highest EFT richness was observed in the supra- and mesomediterranean, particularly in the southern face of the Sierra (Fig. 5c), where the number of vegetation series


is also greater than in the rest of the bioclimatic floors (Valle et al., 2003). The presence EFTs hotspots mainly in the mid-mountain, and particularly in the southern face, could be related to two factors. On the one hand, many Mediterranean mountains show high values of beta diversity up to 1750-1800 m (Wilson and Schmida, 1984), when there is an important structural and compositional replacement of their vegetation. On the other hand, in the middle mountain and especially in its southern face, there is a very diverse mosaic of different types of natural


vegetation mixed with different types of reforestation, traditional crops and uses (Camacho et al., 2002), which gives them the characteristic of multifunctional landscapes from the point of view of the provision of ecosystem services (García-Nieto et al., 2013; Mastrangelo et al., 2014; Cabello et al., 2019). Molero Mesa et al., (1996) and Fernández Calzado et al., (2012) indicated that Sierra Nevada species richness decreases with altitude, while endemic taxa increases (Blanca et al., 2019). Something similar can be observed in the functional diversity of


ecosystems, since the maximum richness is found in areas of medium altitude. The areas with the lowest EFT richness were located in the oro and crioromediterranean levels, and in the eastern semi-arid thermomediterranean extreme, where the harsh soil and climatic conditions (Peinado et al., 2019) diminish floristic diversity although their endemicity increases (Fernández Calzado et al., 2012). The lowest values of EFT richness (richness 4-5) were found in the supramediterranean oak groves, (as in Dionisio et al., 2012; Requena-Mullor et al., 2018) maybe


due to the internal homogeneity of their environmental conditions and their floristic composition (Pérez-Luque et al., 2015, Requena-Mullor et al., 2018).

EFT rarity was highest in the crioromediterranean level, overlapping the area with the highest concentration of endemisms (Cañadas et al., 2014; Peñas et al., 2019) (Fig. 5d). Crioromediterranean vegetation develops under a


very particular ecological conditions that determine uncommon types of ecosystem functioning (rarity 0.6; Fig. 5d), such as, for example, in relatively mobile rocks and canchales located on steep slopes, where the percentage of rarity or compositional endemicity rises to 80% (Blanca and Algarra, 2011). EFT rarity was also high in the eastern end of the semi-arid thermomediterranean level, located in the biogeographic sector of Almeria (Peñas et



al., 2019) with a high concentration of endemisms typical of the Desert of Tabernas (Mota et al., 2004). In the oromediterranean, EFT rarity decreased and reached its minimum, due to the great extension in the Sierra Nevada of this bioclimatic level, which made its functioning not appear as rare, and increasing again in the supra- and mesomediterranean (Fig. 5d). The most rare supra- and mesomediterranean vegetation types corresponded to coniferous and holm oak repopulations (rarity 0.6). The high rarity of coniferous repopulations may be due to disturbances or management interventions that give rise to unique functions in the different masses of conifers.

On the other hand, the rarity in holm oaks may be due to their exclusive functioning, i.e. they have very specific associated EFTs (e.g., Cc1, Dc1). However, the rarity of the different vegetation types (between 0.45 and 0.64) was far from the maximum possible (1).

**4 Data applications for research and conservation / Example of data usage**

Ecological research based on spectral vegetation indices plays an important role in biodiversity conservation (Cabello et al., 2012; Pettorelli, 2016, 2018) and management (Pelkey et al., 2003; Cabello et al., 2016) and for the study of biodiversity and ecosystems responses to environmental changes (Alcaraz-Segura et al., 2017; Pérez-Luque et al. 2015). In fact, numerous studies have demonstrated the usefulness of satellite image time series to evaluate the functional changes in ecosystems at regional scale (Alcaraz-Segura et al., 2010) and at the protected

area level (Alcaraz-Segura et al., 2009; Lourenço et al., 2018). Recently, the use of EFAs derived from spectral indices of vegetation in species distribution models, has made it possible to evaluate with great spatial and temporal precision the suitability of habitat for plant species (Arenas-Castro et al., 2018) and animals (Requena-Mullor et al., 2017; Regos et al., 2019) and may even anticipate expected changes in the distribution of plant species threatened as a consequence of climate change (Alcaraz-Segura et al., 2017). In addition, based on the

EFAs, a monitoring programme of the Spanish National Parks Network has been designed to identify changes and anomalies in functioning, informing managers of the health and conservation status of ecosystems (Cabello et al., 2016).

Furthermore, EFTs  have been used to characterize spatial and temporal heterogeneity of ecosystem functioning

at local and regional scales (Fernández et al., 2010; Cabello et al., 2013); to describe biogeographical patterns (Alcaraz-Segura et al., 2006; Ivits et al., 2013); to evaluate the environmental and human controls of ecosystem functional diversity (Alcaraz-Segura et al., 2013);  to identify priorities for Biodiversity Conservation (Cazorla et al., 2019); to assess the representativeness environmental networks (Villarreal et al., 2018); to assess the effects of land-use changes on ecosystem functioning (Oki et al., 2013); or to improve weather forecast models (Lee et

al., 2013; Müller et al., 2014).

The data sets that we are providing give to the scientific community valuable information of the first characterization of the functional diversity at ecosystem level developed in the entire protected area. We provided a detailed characterization of the functional diversity at ecosystem level for Sierra Nevada, that could be useful to

monitor the response of ecosystems to global change and management actions, to understand the ecosystem functioning and functional diversity across the environmental gradients at protected area scale, and to evaluate the role of ecosystems in providing ecosystem services (Cabello et al., 2019). Sierra Nevada is also a long-term ecological laboratory established 10 years ago (Zamora et al. 2016, 2017), that have available data on biodiversity, climate, ecosystem services, hydrology, land-use changes and management practices from Sierra Nevada. This

will allow to explore the relationships between these other environmental data with the ecosystem functional data that we provide.

**5 Data availability**


The datasets described in this article are available in open-access sources. To broaden their use, first, we provide data in .tif format. Second, we have incorporated rendered versions of all layers as required by Google Earth Pro (called "filename..._forGoogleEarthVisualization.tif") for visualization. And third, we have also developed an ad-hoc visualization platform for all the layers. Datasets available for download in PANGAEA:



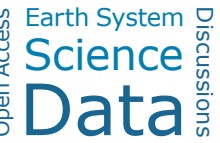

https://doi.pangaea.de/10.1594/PANGAEA.904575 (Cazorla et al. 2019) and for visualisation in
       http://obsnev.es/apps/efts_SN.html.

       The MODIS database used in this work are maintained by NASA (satellite Terra, sensor MODIS, product
       MOD13Q1.006), and the geospatial datasets of Sierra Nevada Park are included in public database of the
Andalusian regional government (REDIAM).

**6 Conclusion**

This dataset provides a characterization of ecosystem functioning and ecosystem functional diversity in Sierra
       Nevada Biosphere Reserve (SE Spain) through the analysis of time series of satellite images of spectral vegetation
       indices as surrogates of the carbon gains dynamics. First, three Ecosystem Functional Attributes (EFAs) describe
       the spatial and inter-annual variability in productivity, seasonality and phenology of vegetation photosynthetic
       activity. Second, the combination of these EFAs into a synthetic classification, i.e. Ecosystem Functional Types
(EFTs), integrates in a single map the spatial heterogeneity of these descriptors of the seasonal dynamics of carbon
       gains. Finally, by using EFTs as biological entities, the spatial patterns of ecosystem functional diversity were
       assessed by means of EFT richness and EFT rarity, as well as the inter-annual variability in ecosystem functioning
       through EFT inter-annual variability and EFT inter-annual dissimilarity.

Ecosystem Functional Types approach improve the understanding of ecosystem processes through environmental
       gradients and provide both the scientific community with valuable information of the first characterization of the
       functional diversity at ecosystem level developed in the entire protected area.

**Author contributions**


       DAS, JC, JP and BPC designed the study, and DAS, JC, JP coordinated it. BPC, AR and EG processed data and
       produced the associated data sets presented in this paper. BPC prepared the manuscript with contributions from
       all authors. BPC and EG prepared the final figures. AJPL design and made the application to visualize the data.

**Competing interests.** The authors declare that they have no conflict of interest.

       **Acknowledgements.** This study was supported by Plan Propio program PhD of University of Almería, it was also
       developed as part of the H2020 project "ECOPOTENTIAL: Improving future ecosystem benefits through earth
       observations" (http://www.ecopotential-project.eu/), which has received funding from the European Union's
       Horizon 2020 research and innovation programme under grant agreement No 641762; and project LIFE-
ADAPTAMED (LIFE14 CCA/ES/000612): "Protection of key ecosystem services by adaptive management of
       Climate Change endangered Mediterranean socio-ecosystems". E.G. was supported by the Spanish Ministry of
       Science under the project TIN2017-89517-P.

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

# Figures

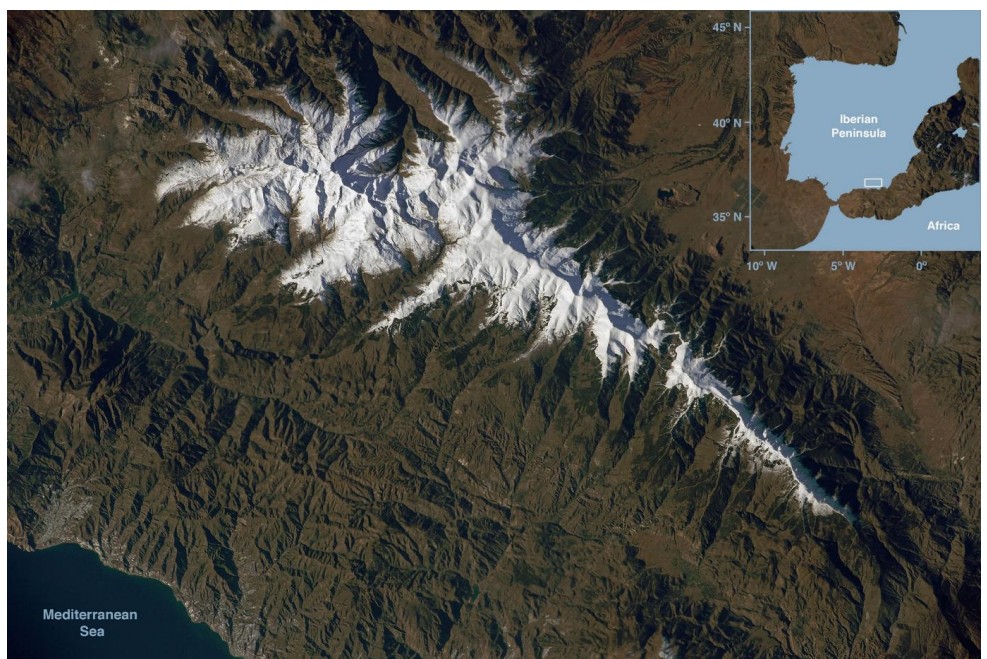

**Figure 1. Location (top-right) and remote view of Sierra Nevada mountain region (image from the International Space Station took in December 2014; courtesy of "Earth Science and Remote Sensing Unit, NASA Johnson Space Center").**


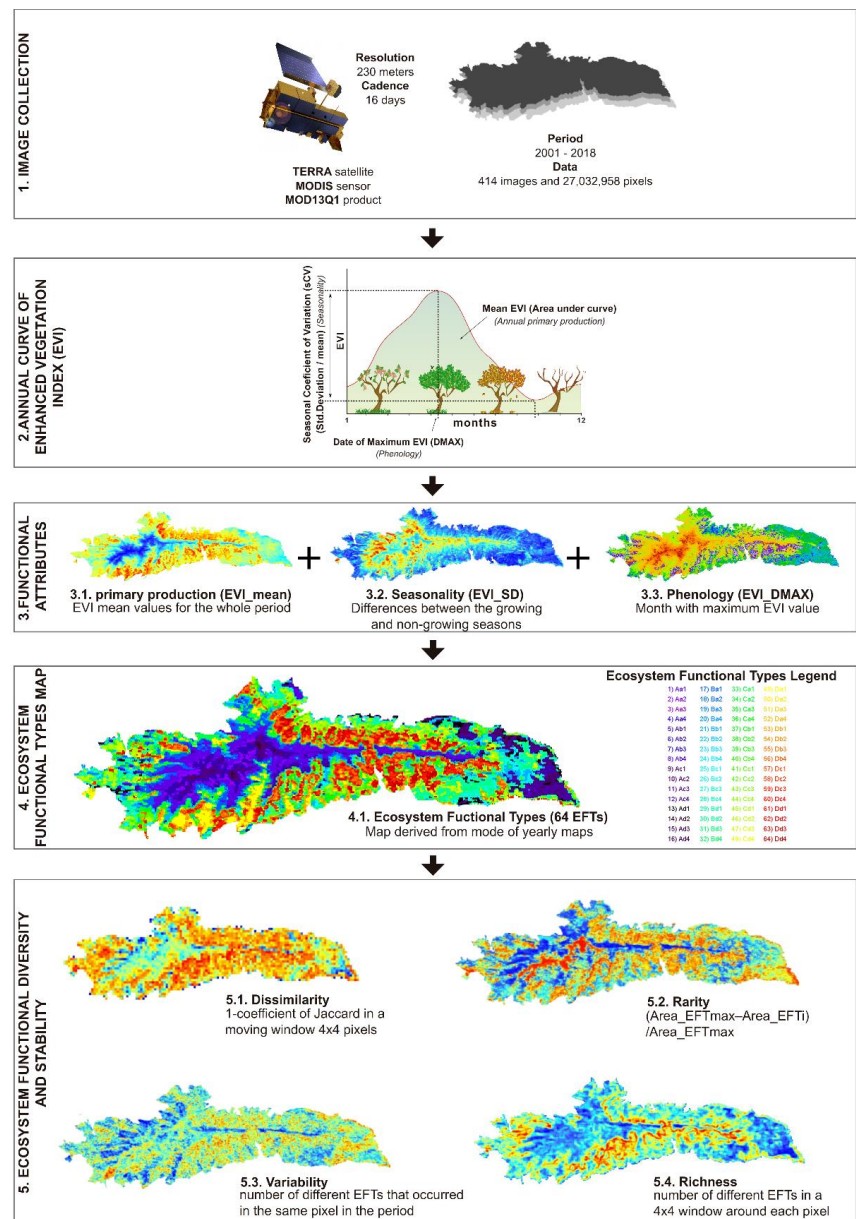

**Figure 2. Workflow to characterize the ecosystem functioning and functional diversity of Sierra Nevada. MODIS (Moderate Resolution Imaging Spectroradiometer) sensor product MOD13Q1 was used aboard NASA's Terra satellite. This product contains images with 16-day temporal resolution (23 images per year) and ~232 m spatial resolution from the Enhanced Vegetation Index (EVI). The study period was from 2001 to 2018. Three functional attributes describing ecosystem functioning were calculated from the EVI seasonal curve for each year. The range of values for each attribute was divided into four intervals, resulting in a potential number of 64 TFEs (4x4x4=64). From EFTs, we derived fourth metric related to ecosystem functional diversity (EFT richness and rarity) and ecosystem functional stability (interannual variability and dissimilarity).**

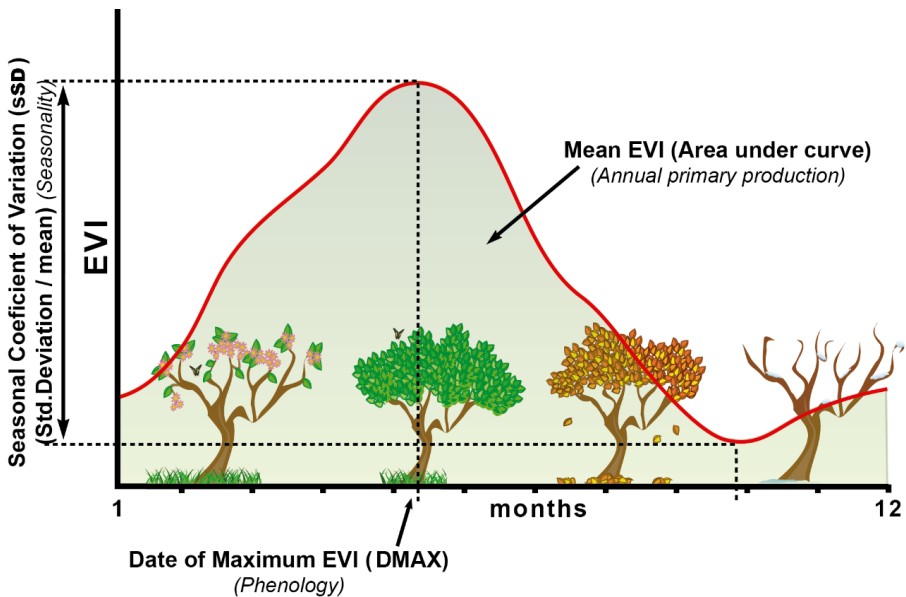

**Figure 3: Seasonal dynamics of Enhanced Vegetation Index (EVI) and EVI derived metrics or Ecosystem Functional Attributes (EFAs). The axis "x" corresponds with months and the axis y with EVI values. EFAs were: the annual mean or the area under curve, an estimator of annual productivity (EVI_mean), the EVI seasonal coefficient of variation, i.e. the differences between the minimum and the maximum EVI values, a descriptor of seasonality (EVI_sSD), and the date of maximum EVI, an indicator of phenology (EVI_DMAX). We chose this three EVI metrics or EFAs due to they capture most of the variance of the EVI time series**





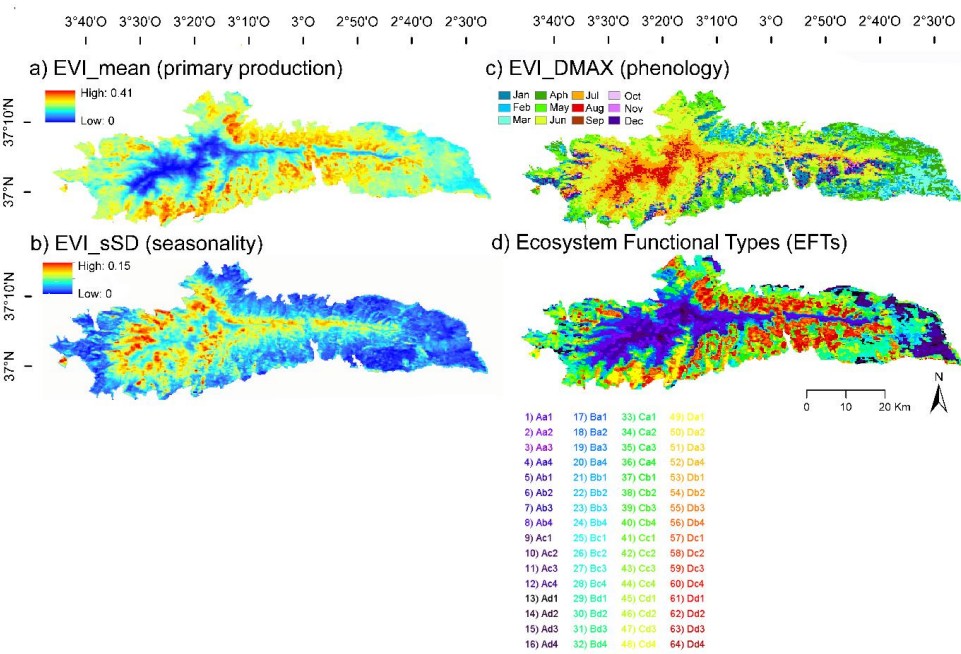

**Figure 4. Ecosystem Functional Attributes (a-c) and Ecosystem Functional Types (d) describing the functioning of the canopy based on the Enhanced Vegetation Index (EVI), derived from MOD13Q1-TERRA (pixel ~232 m) for the period 2001-2018.**

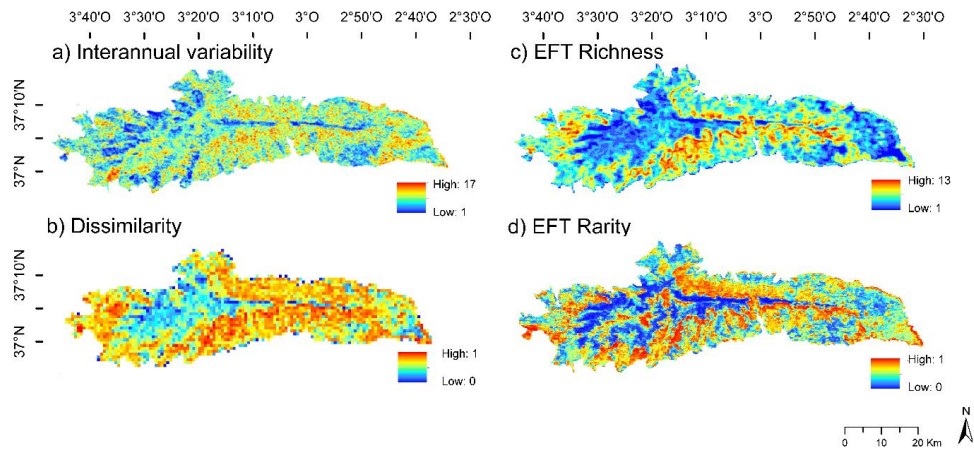


**Figure 5. Functional diversity patterns based on the Enhanced Vegetation Index (EVI), derived from MOD13Q1-TERRA for the period 2001-2018. a) EFTs interannual variability for the period; b) EFTs interannual dissimilarity or 1 - Jaccard coefficient for the period; c) Spatial EFT richness patterns from a 4x4 pixel MODIS mobile window (∼1 km2); and d) Spatial EFT rarity patterns.**




**Table 1. EFAs range used for identification of EFTs in Sierra Nevada. For EVI_DMAX, the four intervals agreed with the four seasons of the year. For EVI_mean and EVI_sSD, we extracted the first, second, and third quartiles for each year and then calculated the inter-annual mean of each quartile (means of the 18-year period).**


| Ecosystem Functional Attribute | Character code | Digit code | Range |
|---|---|---|---|
| EVI Mean (Productivity) | A | 100 | 0 - 0,137 |
| | B | 200 | 0,137 - 0,187 |
| | C | 300 | 0,187 – 0,241 |
| | D | 400 | > 0,241 |
| EVI SD (Seasonality) | a | 10 | > 0,062 |
| | b | 20 | 0,043 – 0,062 |
| | c | 30 | 0,030 – 0,043 |
| | d | 40 | 0 – 0,030 |
| EVI MMAX (Phenology) | 1 | 1 | Spring |
| | 2 | 2 | Summer |
| | 3 | 3 | Autumn |
| | 4 | 4 | Winter |





**Table 2. Dataset description:** Ecosystem Functional Attributes (EVI_Mean, EVI_sSD and EVI_MMAX provided yearly and summarized for the period); Ecosystem Functional Types (EFTs yearly and summarized for the period (mode, interannual variability and dissimilarity); Ecosystem Functional Diversity (EFT richness and EFT rarity, provided yearly and summarized for the period). Spatial resolution is ~232 in all cases except in the EFT dissimilarity, where it is ~298. YYYY refers to year and varies from 2001 to 2018.

| Filename | Variable | Definition | Biological significance | Temporal resolution |
|---|---|---|---|---|
| EVI_Mean_YYYY_C006_MOD13Q1_Pixel232 | EVI_mean | Mean of the positive EVI values in a year | Primary production in a year | Yearly, one image per year YYYY |
| EVI_Mean_InterAnnualMean_2001-2018_C006_MOD13Q1_Pixel232 | EVI_mean | Inter-annual mean of the annual EVI_mean values of the period | Average annual primary production of the period | One image for the 2001-2018 period |
| EVI_sSD_YYYY_C006_MOD13Q1_Pixel232 | EVI_sSD | Intra-annual standard deviation of the positive EVI values within a year | Seasonality in vegetation greenness. Differences in carbon gains between the growing and non-growing seasons in a year | Yearly, one image per year YYYY |
| EVI_sSD_InterannualMean_2001-2018_C006_MOD13Q1_Pixel232 | EVI_sSD | Inter-annual mean of the annual EVI_sSD values of a period | Seasonality. Average annual of the differences in carbon gains between the growing and non-growing seasons throughout the period | Average of the 2001-2018 period |
| EVI_MMAX_YYYY_C006_MOD13Q1_Pixel232 | EVI_MMAX | Month with maximum EVI in a year | Phenology. Date of maximum greenness in a year | Yearly, one image per year YYYY |
| EVI_MMAX_InterannualMean_2001-2018_C006_MOD13Q1_Pixel232 | EVI_MMAX | Inter-annual mean of the month with maximum EVI of the period | Phenology. Average annual of the month with maximum greenness throughout the period | Average of the 2001-2018 period |
| EFTs_YYYY_C006_MOD13Q1_Pixel232 | EFTs | Range of EFA's values divided into four intervals $4 \times 4 \times 4 = 64$ potential EFTs in a year | Patches of land surface that share similar dynamics in matter and energy exchanges in a year | Yearly, one image per year YYYY |






| Variable | Name | Formula | Description | Temporal |
|---|---|---|---|---|
| EFTs_InterannualMode_2001-2018_C006_MOD13Q1_Pixel232 | EFTs | Mode of the range of EFA's values divided into four intervals $4 \times 4 \times 4 = 64$ potential EFTs of the period | Patches of land surface that share similar dynamics in matter and energy exchanges throughout the period | Mode of the 2001-2018 period |
| EFT_InterannualVariability_2001-2018_C006_MOD13Q1_Pixel232 | EFT interannual variability | N° of different EFTs that occurred in the same pixel in the period | Changes in an ecosystem functioning in a period | 2001-2018 period |
| EFT_InterannualDissimilarity_2001-2018_C006_MOD13Q1_Pixel232 | EFT interannual dissimilarity | 1 - *Jaccard Index* | Changes in ecosystem functioning a landscape level in a period | 2001-2018 period |
| EFT_Richness_YYYY_C006_MOD13Q1_Pixel232 | EFT richness | N° of different EFTs in a $4 \times 4$-pixel moving window around each pixel in a year | Different EFTs represented in the land-surface in a year | Yearly, one image per year YYYY |
| EFT_Richness_InterannualMean_2001-2018_C006_MOD13Q1_Pixel232 | EFT richness | N° of different EFTs in a $4 \times 4$-pixel moving window (924 x 924 m) around each pixel in a period | Different EFTs represented in the land surface throughout the period | Average of the 2001-2018 period |
| EFT_Rarity_YYYY_C006_MOD13Q1_Pixel232 | EFT rarity | *Rarity of EFTi = (Area_EFTmax–Area_EFTi)/Area_EFTmax (in a year)* | EFT geographical extension | Yearly, one image per year YYYY |
| EFT_Rarity_InterannualMean_2001-2018_C006_MOD13Q1_Pixel232 | EFT rarity | *Rarity of EFTi = (Area_EFTmax–Area_EFTi)/Area_EFTmax (in a period)* | EFT geographical extension | Average of the 2001-2018 period |