# Peer review of "A remote sensing-based dataset to characterize the ecosystem functioning and functional diversity of a Biosphere Reserve: Sierra Nevada (SE Spain)"

_Earth System Science Data, 2019_

## Referee Comment (RC1) · Anonymous Referee #1 · 23 Mar 2020

General comments:
The authors provide a valuable compilation of remote sensing based indicators that that are used to characterize the ecosystem condition of a test site in south-eastern Spain (Sierra Nevada). The indicators are computed from time series of Enhanced Vegetation Index (EVI) data from 16-day MODIS maximum value composite (MVC) data. The framework for the assessment of ecosystem functioning and functional diversity builds on a set of temporal metrics that are computed on an inter-annual, annual or seasonal level as well as on metrics that capture the spatial heterogeneity

of the derived metrics. These metrics are used as proxies for ecosystem functional attributes (EFAs). The analysis of the temporal variability of the EFAs yielded the ecosystem functional types (EFTs) and the spatio-temporal heterogeneity of EFTs resulted in the characterization of ecosystem functional diversity. The main rationale behind this framework is that ecosystem primary production can be assessed from satellite vegetation indices and that primary production is the key indicator of ecosystem functioning. Overall, the proposed framework for computing EFTs and functional diversity from satellite time series is comprehensible and well documented. The translation of temporal metrics of vegetation indices into functional attributes and type of ecosystems is well-founded and presents a prototype for large-scale ecosystem assessment and monitoring. The description of the datasets is appropriate and the data are available, structured and labelled logically. However, there are a few specific issues that need to be addressed before the manuscript can be accepted for publication:

Specific comments:
1. The authors do not provide any information about the processing of the MODIS13Q1.006 time series to annual and inter-annual image stacks. Here, the most important point that has to be considered is the masking of valid pixels (clouds, aerosols, snow / ice, etc.) based on the quality assessment (QA) layer (VI Quality) of the MODIS dataset. The clarification on this issue is crucial has a direct impact on a number of the more technical comments below.
2. If no masking has been carried out, the whole results section has to be revised.
3. The study area is rather small (2000 km2) and the landscape in the area shows small-scaled patterns of land-use patches. Why did you use coarse scale satellite data for your analysis and not the archive of available medium resolution satellite data (e.g. Landsat) for your work? This is more a general question, I do not really expect that you redo the full work, however, you could add a conclusive remark at the end of your work.

Technical comments:

Line 126: explain EVI and add reference

Line 130: linked site is not available

Line 130/131: the doi is not related to GEE. Please adjust accordingly either the link or the description.

Line 131/132: EVI between 0 and 1000 – in tables you use scaling from 0-1; what about negative values? The full data range is from -1 to +1.

Line 135/136: How did you identify these 3 metrics? There are a number of additional phenological metrics available that are known to represent meaningful features of ecosystem productivity (e.g. start / end, length of season). What is "biologically meaningful" in the context of your research?

Line 139/140: How did you define the growing season?

Line 147/148: I doubt that you will have EVImax in the winter period after clearing your EVI data for snow/ice, clouds, etc.

Line 161: "relative extension" - what do you mean, here? Share of area of EFTi within a defined area (moving window)?

Line 162: "compared to the most abundant EFT" – in a defined area / window?

Line 218: "altitudinal patterns"- What about topographical patterns (aspect, slope)?

Line 219 ff.: I cannot find any map of those bioclimatic belts for the study area. Hence, I am not able to follow the description of results. Please add a figure.

Line 235: "maximum greenness in winter" – see comment above, how would you explain a greenness peak in wintertime?

Line 254: "interannual variability ranged from 1 to 17 different EFTs over the 18-year period" - what is the contribution of data uncertainty / data quality in this context, e.g. the missing QA-masking on one side and the very low EVI values on the other hand?

Line 359: "geospatial data Sierra Nevada Park" – Where do you show these data?

Line 366: "Sierra Nevada Biosphere Reserve (SE Spain)" – show in map!

Figure 1: It would be helpful for the interpretation of the EFA and EFT data to have a map of vegetation types rather than a simple snapshot from the ISS without any

information on content and scale.

Figure 3: the mean EVI is NOT the "area under curve"! This would rather be the cumulative EVI.

[Figure]

---

## Referee Comment (RC2) · Anonymous Referee #2 · 26 Mar 2020

Are the data and methods presented new?  -An interessting approach is presented for inter-annual heterogeneity; it is left open why for assessing the spatial variability a certain kernel size had ben chosen

Is there any potential of the data being useful in the future? -In principle yes, however, there are details missing, see next

Are methods and materials described in sufficient detail? - No. Why is the kernel size 4x4?  How have borderline pixels be processed with the kernel?  (kernel processed

raster layer have same extension) How variable are the quartile boundaries (could you name a standard deviation?)

Are any references/citations to other data sets or articles missing or inappropriate? -reference/URL to the database REDIAM is missing, also, which particular datasets have been employed from it; by what data got the MODIS data clipped/masked?

Is the article itself appropriate to support the publication of a data set? - yes with respect to gain an understanding of the data. The article does not provide necessary information to re-use the data: the legend for EFTs is part of Fig 2; the values of the EFTs do not correspond to the values in the TIFs (there they are 1-64 encoded)

Check the data quality: Is the data set accessible via the given identifier? -yes Is the data set complete? -yes Are error estimates and sources of errors given (and discussed in the article)? - well, not error but there is no reference to variability eg the means of internal quartiles given Are the accuracy, calibration, processing, etc. state of the art? - The article employs community-"standard" pre-processed data; however, it does not provide accuracy information of intermediate processing steps. Also, the derivation of spatial heterogeneity, the chosen size of the kernel and how this affects the results is not discussed

Are common standards used for comparison? - the resulting data are not compared Is the data set significant – unique, useful, and complete? -The data set is useful

Consider article and data set: Are there any inconsistencies within these, implausible assertions or data, or noticeable problems which would suggest the data are erroneous (or worse). - using a kernel to derive values I would have expected that the resulting layer is smaller in size than the input layer, unless some "mirroring" is done to extend the input layer in size. The article does not provide any information on how this was handled

If possible, apply tests (e.g. statistics). - looking up the TIFs with standard GIS software

(QGis) did not reveal any problems. The histograms of values seem ok, although because of missing legend they could not be really interpreted

Is the data set itself of high quality? Check the presentation quality: Is the data set usable in its current format and size? -yes, the GeoTIFF is a well accepted and documented file format

Are the formal metadata appropriate? - No, I am unable to discover any formal metadata. The GeoTIFF come with some metadata in their header, but do require specialized software for extraction, eg. of the bounding box or employed projection. additional TFW file would be readable with common editors. Additional formal metadata is missing.

Check the publication: Is the length of the article appropriate? - given, that it is a data publication, the article dwells much on discussion of the application/biodiversity/structure but is much shorter when it comes to describing data and methodology

Is the overall structure of the article well structured and clear? -yes

Is the language consistent and precise? -there are a few language errors but the article is language wise in good shape

Are mathematical formulae, symbols, abbreviations, and units correctly defined and used? - Eq.3 uses X any Y without explicit definition; this equation does not provide additional information content Are figures and tables correct and of high quality? Quality is mostly acceptible, in Fig.2, part 3 the legend is hardly readable

Finally: By reading the article and downloading the data set, would you be able to understand and (re-)use the data set in the future? -No, eg. the EFT type as encoded in the TIFs cannot be interpreted

Uniqueness: It should not be possible to replicate the experiment or observation on a routine basis. - all resulting data can be reproduced as the primary source is generally

available However, the derivation needs expertise with GIS/remote sensing software, and a target audience of ecologists is usually easier reached with data products which are deemed useful for such clientele

The introduced methods are not trivial nor obvious, however, would benefit from a descussion why certain approaches had been taken (kernel size eg.) The data seem complete. All derived data sets are provided (annual data), also the summary data. In theory one could re-calculate all results (if eg. interval boundaries were to be now known, EVI_max).

I would request information on hardware and software used to derive products (algorithmic deviations) Also, to reproduce the data information on masking/clipping the covered regions is necessary but absent. (which dataset, which method)

Technical details:

line24: imagery do not provide a continous characterization as reflectance is integrated per pixel

line 26: from 2001 to 2018

line 79 not the EFT approach has exp. grown but the application of EFT approaches

line 137 EFT seasonal curve: the term has not been introduced properly; I presume it refers to the 23 measurements taken per year, please clarify

line 146: one cannot understand the present derivation as the methodology is refered to another article; worse, the authors write of a "similar" approach without making clear how/where they differ

line 147 EVI_DMAX: unclear, whether you chose the intervals according to the definition of the seasons or you derived them and they turned out to coincide with the seasons; please clarify

line 149-150: the derivation of quartile borders was understandable only after consulting the reference. How stable are the boundaries, that is, provide a standard deviation for each mean Table 1: values cannot be reproduced nor checked, e.g. EVI_Mean_2001_C006_MOD13Q1_Pixel232.tif shows values between 11.5-4471.9 (QGis), table 1 reports 75% values are less than 0.241 EVI_mean: problem with "sealed" class boundaries: derivation relies on mean of a 18y period. If say, you want to show the time series of 2001-2020, would you need to do the derivation of the boundaries or "extrapolate" from 2018? Table 1, EVI_Max: values of 1-4 do not correspond to values found in TIFs (1-12)

line 159: justification for a 4x4 kernel? Why not 3x3 or 5x5? Could the kernel be dependend on the question being asked? How have borderline pixels be processed/why eg share richness and inter-annual mode the same borders?

line 359: database is maintained

line 360: please include a reference/URL to the database REDIAM, also, indicate which datasets of REDIAM have been included in your work

Fig 2.1; https://lpdaac.usgs.gov/products/mod13q1v006/ states 250m GSD, not 230m. Fig 2.2: the mean is not the area under the curve, but the area normalized by the range; there is no curve at all but 23 discrete values/year Fig 2.4: the legend is crucial for re-using data but is not provided as individial data (eg. numerical values corresponding to a class, or pseudo color code for GoogleEarth); at present, the TIF files for eg EFTs show values between 1-64; how to map to your classes?

---

## Author Comment (AC1) · 10 Apr 2020

Dear Reviewer,

Many thanks for your correspondence regarding our data description paper entitled "A remote sensing-based dataset to characterize the ecosystem functioning and functional diversity of a Biosphere Reserve: Sierra Nevada (SE Spain)". We thank you for all your constructive comments, which provided valuable insights to improve the conceptual and methodological robustness of our data and our paper. We are now very

pleased to send you the response to your comments and suggestions.

In our response below, please find our point-by-point responses (indicted with "R") presenting, in detail, how we have addressed the Reviewer comments ("C"). In the .pdf document attached, the Reviewer comments are reproduced in bold italic font and our responses are indicated in plain text, in addition, tables and figures are embed in the main document. We numbered each comment and reply for ease of reference and indicated changes that will be made in the manuscript.

Once again, we thank you for your time, constructive comments and suggestions. We hope to meet the expectations with this response, and that the Reviewer considers our data description manuscript suitable to be published in Earth System Science Data.

Sincerely,

The authors

Referee #1 General comments: The authors provide a valuable compilation of remote sensing based indicators that are used to characterize the ecosystem condition of a test site in south-eastern Spain (Sierra Nevada). The indicators are computed from time series of Enhanced Vegetation Index (EVI) data from 16-day MODIS maximum value composite (MVC) data. The framework for the assessment of ecosystem functioning and functional diversity builds on a set of temporal metrics that are computed on an inter-annual, annual or seasonal level as well as on metrics that capture the spatial heterogeneity of the derived metrics. These metrics are used as proxies for ecosystem functional attributes (EFAs). The analysis of the temporal variability of the EFAs yielded the ecosystem functional types (EFTs) and the spatio-temporal heterogeneity of EFTs resulted in the characterization of ecosystem functional diversity. The main rationale behind this framework is that ecosystem primary production can be assessed from satellite vegetation indices and that primary production is the key indicator of ecosystem functioning. Overall, the proposed framework for computing EFTs and functional diversity from satellite time

series is comprehensible and well documented. The translation of temporal metrics of vegetation indices into functional attributes and type of ecosystems is well-founded and presents a prototype for large-scale ecosystem assessment and monitoring. The description of the datasets is appropriate and the data are available, structured and labelled logically. However, there are a few specific issues that need to be addressed before the manuscript can be accepted for publication:

Specific comments:

C1. - * The authors do not provide any information about the processing of the MODIS13Q1.006 time series to annual and inter-annual image stacks. Here, the most important point that has to be considered is the masking of valid pixels (clouds, aerosols, snow / ice, etc.) based on the quality assessment (QA) layer (VI Quality) of the MODIS dataset. The clarification on this issue is crucial has a direct impact on a number of the more technical comments below.

R1. - Thank you very much for raising this question. We used Enhanced Vegetation Index (EVI) since it minimizes canopy background variations and maintains sensitivity over dense vegetation conditions (Liu and Huete 1995). The MODIS EVI also uses the blue band to reduce residual atmosphere contamination caused by smoke and sub-pixel thin clouds (Huete et al. 1999). The MODIS EVI products are computed from atmospherically corrected bi-directional surface reflectances. Furthermore, the algorithm used by this product (MOD13Q1.006 product) chooses the best available pixel value from all the acquisitions from the 16 day period. The algorithm operates on a per-pixel basis and requires multiple observations (16 days) to generate a composited EVI. Due to orbit overlap, multiple observations may exist for one day and a maximum of four observations per day may be collected. The MOD13Q1 algorithm separates all observations by their orbits providing a means to further filter the input data.

Once all 16 days are collected, the MODIS algorithm applies a filter to the data based on quality, cloud presence, and viewing geometry (Fig. 1). Cloud-contaminated pixels

and extreme off-nadir sensor views are considered lower quality. A cloud-free, nadir view pixel with no residual atmospheric contamination represents the best quality pixel. Only the highest quality, cloud-free, filtered data are retained for compositing (Huete et al. 1999, Didan 2015b). The goal of the compositing methodology is to extract a single value per pixel from all the retained filtered data, which is representative of each pixel over the particular 16-day period. The compositing technique uses an enhanced criteria for normal-to-ideal observations, but switches to an optional backup method when conditions are less than ideal (Fig. 1).

The EVI values range from -1 to +1, where negative values generally correspond to snow, ice, or water; and values closer to +1 represent the higher density of green leaves (Huete et al. 2002). In our data, in addition to assuming the correct native pre-processing of the data explained above, negative values (associated with snow, ice or water) were transformed into zeros.

Despite the high standard quality of the 16-day EVI maximum value composite in MOD13Q1, we have assessed the effect of the additional application of the QA mask flags on the three Ecosystem Functional Attributes that are used as the basis for our further analyses and maps (e.g. the three metrics: EVI_mean, EVI_SD and EVI_DMAX). To do this, we have calculated EFAs using the "Summary Quality Assessment" band of MOD13Q1 product and masking out (values were substituted by NANs) the values 2: pixel covered with snow/ice, and 3: pixel cloudy. For EVI_mean and EVI_SD (continuous variables), and we carried out two comparisons: 1) we calculated a simple sliding window (3x3 pixels) correlation (Pearson correlation) between two rasters (data with QA mask and our data with negative values transformed into zeros), and plotted the map and histogram of the correlation coefficients (Fig. 2a-d), and 2)we calculated the linear regression between the filtered and unfiltered EFAs for the average year (Fig. 3). Most pixels had correlation values greater than 0.9, and the small areas with lower correlation between the filtered and non-filtered EVI_SD mainly occurred in oromediterranean belt, above the treeline. For EVI_DMAX, we assessed

the impact of masking by subtracting the filtered and unfiltered EVI_DMAX layers, once classified into seasons, to map the pixels that changed (mapped as 0) and those that did not change (mapped as 1) and to produce the corresponding histogram (Fig. 2e,f). Here, we only observed a small percentage of pixels with changes (4.35% of pixels changed and 95.65% did not change the EVI_DMAX season), located mainly in the oro- and crioromediterranean belts and the changes were from spring (with filtering using the QA mask) to summer (without QA mask). Therefore, the functional attribute less affected by the filtering was EVI_Mean, the one that has more weight in our data, the surrogate for primary production (see R8).

We also calculated linear regressions between the filtered and non-filtered, detecting a high relationship between both for EVI_mean (R=0.99), with slightly higher values of EVI for QA data (Fig. 3a) and a little more dispersion for EVI_sd (R=0.97), with lower values of sd for QA data (Fig. 3b).

Considering the small effect of filtering using the Quality Assessment bands on Ecosystem Functional Attributes and the very time-consuming effort that represents reprocessing all data, we have decided not to filter the dataset so far. However, if the reviewer and editor still think that we should apply the QA filtering, we will filter out snow, ice and water as zeros and clouds as NANs.

In case that the editor considers that no filtering is required, we would add the following text in the manuscript into section 2.2.

"MOD13Q1.006 EVI product is computed from atmospherically corrected bi-directional surface reflectances by choosing the best available pixel value from all the acquisitions (4 per day) in a 16 day period based on quality, cloud presence, and viewing geometry (Huete et al., 1999, Didan et al. 2015a). In addition, to further remove the potential remaining effect of snow, ice and water in our dataset, negative EVI values were transformed into zeros."

C2. - * If no masking has been carried out, the whole results section has to be revised.

R2. - Please, see response R1. We will provide comments (e.g. advantages and limitations) on this topic in the Discussion section of the manuscript after the open online discussion.

C3. - * The study area is rather small (2000 km2) and the landscape in the area shows small-scaled patterns of land-use patches. Why did you use coarse scale satellite data for your analysis and not the archive of available medium resolution satellite data (e.g. Landsat) for your work? This is more a general question, I do not really expect that you redo the full work, however, you could add a conclusive remark at the end of your work.

R3. - Thank you very much for your interesting question. Using MODIS instead of other satellites with higher spatial resolution (e.g. Landsat) has several advantages in terms of data quality (e.g. presence of clouds) along the time series. Since the MODIS sensor provides a daily image of the Earth, such high frequency (1 per day) increases the probability of finding a cloud-free image every 16-days (see response R1). MODIS provides the best composite value every 16 days (i.e. chooses the best available pixel value from all the acquisitions from the 16 day period), applying an algorithm that selects the image atmospherically corrected bi-directional surface reflectances and select the image with lowest cloud presence, the lowest view angle, and the highest EVI value (see response R1). Although Landsat has a lower pixel size, their images have a lower frequency (i.e., 1 image every 16 days). Thus, the fixed acquisition schedule makes it less probable to acquire good-quality imagery for a particular place periodically (especially if clouds occur frequently over the area of interest, e.g. winter dates).

Second, the Landsat product with the largest time series is Landsat 7 (1999-actually), however, on May 31, 2003, the satellite's scan-line corrector failed. The scan-line corrector is a device on the satellite that keeps the scan lines parallel to each other. Without the Scan Line Corrector (SLC), the scan lines are mis-aligned and there are wedge-shaped data gaps in the image (see sample Fig. 4 for Sierra Nevada). Providers offer different procedures for filling-in the data gaps, but each amounts to using data from good images prior to 2003 to do so. Obviously the further one gets from 2003,

the less valid this approach will be. Therefore, since 2003 SLC failure of Landsat 7, Landsat 8 is the only fully operational Landsat satellite in orbita, but covers a shorter time series than MODIS (Landsat8 covers from 2013 to actually, while MODIS covers from 2001 to actually).

Other satellites have also been considered for their use, as Sentinel, which also has a higher spatial resolution but the time series is still too short for long-term assessments (2014-present).

Finally, we consider appropriate MODIS spatial resolution for ecological studies at protected-area level, according to Anderson (2018), which showed that the temporal resolution of MODIS is useful for characterizing the seasonal dynamics of ecosystem functioning (Fig. 5). Furthermore, there are other works that use MODIS successfully at protected-area level (e.g. Lourenço et al. 2018, Requena-Mullor et al. 2018).

The arguments for the choice of MODIS were not included in the manuscript, therefore we would add the following sentence that briefly justify the choice of MODIS (section 2.2.): "Despite its moderate spatial resolution (aprox. 230 m/pixel), we chose MODIS since it offers a long time series (almost 20 years) of almost cloud-free images every 16 days thanks to the maximum value composite of daily images,which allows for the characterization of the temporal dynamics of ecosystem functioning (Anderson et al. 2018)".

Furthermore, we will also include comments in the Discussion section on the potential of extending our approach to Sentinel-2 data once the time-series gets longer.

Technical comments:

C4. - * Line 126: explain EVI and add reference

R4. - We chose EVI instead of any other vegetation index (such as SAVI, ARVI, or NDVI) as an indicator of carbon gains since it is supposed to be more reliable in both low and high vegetation cover situations (Huete et al. 1997). EVI is sensitive

to changes in areas having high biomass, EVI reduces the influence of atmospheric conditions on vegetation index values, and EVI corrects for canopy background signals (see R1).

EVI is computed following this equation:

EVI=G(NIR-red)/ (NIR+C1*red-C2*blue+L),

where NIR/red/blue are atmospherically-corrected (Rayleigh and ozone absorption) surface reflectances, L is the canopy background adjustment that addresses non-linear, differential NIR and red radiant transfer through a canopy, and C1, C2 are the coefficients of the aerosol resistance term, which uses the blue band to correct for aerosol influences in the red band. The coefficients adopted in the MODIS-EVI algorithm are; L=1, C1 = 6, C2 = 7.5, and G (gain factor) = 2.5.

We will explain EVI with its reference in the manuscript in section 2.2.

C5. - * Line 130: linked site is not available

R5. - Thank you for pointing it out. We will replace the old link with the updated one (https://lpdaac.usgs.gov/products/mod13q1v006/ ) (Didan 2015a).

C6. - * Line 130/131: the doi is not related to GEE. Please adjust accordingly either the link or the description.

R6. - Ok, thank you. We will modify the line by adding the official GEE reference: "Gorelick, N., Hancher, M., Dixon, M., Ilyushchenko, S., Thau, D., & Moore, R. (2017). Google Earth Engine: Planetary-scale geospatial analysis for everyone. Remote Sensing of Environment."

C7. - * Line 131/132: EVI between 0 and 1000 – in tables you use scaling from 0-1; what about negative values? The full data range is from -1 to +1.

R7. - Thank you for your comment. We transformed all negative values into 0, since negative EVI values are known to be related to the presence of snow, ice or water (and

therefore, ecologically, it makes more sense to have an EVI value of zero rather than negative) (Huete et al. 2002). In the new version, we will add an explanation about this in section 2.2 (see R1).

C8. - * Line 135/136: How did you identify these 3 metrics? There are a number of additional phenological metrics available that are known to represent meaningful features of ecosystem productivity (e.g. start / end, length of season). What is "biologically meaningful" in the context of your research?

R8. - Biologically, these three metrics can be interpreted as surrogates (Paruelo et al. 2001, Pettorelli et al. 2005, Alcaraz-Segura et al. 2006) of the total amount and timing (seasonality and phenology) of primary production, on of the most integrative indicators of ecosystem functioning (Virginia and Wall 2001). Statistically, these three metrics are known to be highly correlated with the first two or three axes (and hence capture most of the variance) of a Principal Component Analysis (PCA) carried out on the NDVI or EVI annual dynamics in different regions (Townshend et al. 1985, Paruelo and Lauenroth 1998, Paruelo et al. 2001, Alcaraz-Segura et al. 2006, Alcaraz-Segura et al. 2009, Ivits et al. 2013). To know the statistical meaningfulness of these metrics in Sierra Nevada Biosphere Reserve, we also examined their correlation with the first axes of a PCA run on the EVI annual curve of the average year (12 EVI values, i.e. the interannual means of the the maximum value composites for each month). The first two axes cumulated 96.5% of the variance (PC1 87.3%, PC2 9.2%). The eigenvectors showed that the weights along the months were similar for the first PCA axis (even weights throughout the year), while for the second axis they showed a contrast between winter and summer months (Table 1). This indicated that PC1 can be related to the total or average amount of EVI, and that PC2 can be related to the intra-annual variability of EVI (Fig. 6).

In addition, we explored the correlation between the PCA axis and the EVI metrics (i.e., EFAs). The EVI metrics showed high correlation with the PCA axes. PC1 accounted for most of the total variance in the seasonal dynamics of the EVI (87.3%) and was strongly

correlated with the EVI annual mean (PC1 vs. EVI_Mean r = 0.94). PC2 accounted for 9.2% of the total variance (PC1 and PC2 cumulated 96.5% of total variance) and was related to seasonality and phenology metrics (as in Alcaraz-Segura et al. 2006, 2009) (PC2 vs. EVI_SD r = -0.75; PC2 vs DMAX_Sine = 0.67; PC2-vs DMAX_Cosine = -0.61) (Table 2). To correlate DMAX with the PC axes and keep the continuous nature of the annual period and the relative distance between months (i.e. December is as close to January as July is to June, that is, the distance between December (12) and January (1) is one month, not eleven months), we transformed months into polar coordinates. The entire circumference of a year was divided into 12 portions and each month was equated to an angle (30° for January and 360° for December). DMAX months were therefore characterised by their sine and cosine values.

In summary, PC1 was very highly correlated to EVI_Mean and then can be interpreted as annual primary production. PC2 shows a high contrast in the eigenvector values between winter and summer and is highly correlated with EVI_SD and with the Sine and Cosine components of DMAX, so it can be interpreted as a combination of seasonality (SD) and phenology (DMAX). Mathematically, it could be expressed as follows: PC2 = f( a*SD + b*DMAX_Sine + c*DMAX_Cosine + d + e) (Table 1 and 2), and the r-square of this multiple regression was 0.70.

In addition, the EVI metrics were orthogonal, since the correlation between them was low, so that each EVI metric contributed independently to explain the variance of the EVI time series (Table 3).

Hence, these three EVI metrics are both "biologically and statistically meaningful" since they are linked to essential biodiversity variables such as productivity, seasonality and phenology and capture most of the variability of the EVI annual dynamics, a surrogate of primary production dynamics (Monteith 1972), the most integrative indicator of ecosystem functioning (Virginia and Wall, 2001), and the basis for multiple ecosystem processes and services (Paruelo and others 2016).

We will include a summary of this analysis (PCA and correlations) in the manuscript to better justify the biological and statistical meaningfulness of our EFAs.

C9. - * Line 139/140: How did you define the growing season?

R9. - Here we refer to a conceptual more than a technical definition of the growing season, as the period of the year with greater vegetation activity compared to other periods of the year with lower vegetation activity (de Beurs and Henebry 2010, Henebry and de Beurs 2013).

C10. - * Line 147/148: I doubt that you will have EVImax in the winter period after clearing your EVI data for snow/ice, clouds, etc.

R10. - The values of EVImax that we have found can be explained by the changing conditions of the environmental controls of vegetation growth along Sierra Nevada. In the Mediterranean mountains, both summer drought and winter cold are known to be the limiting factors of vegetation growth (Alcaraz-Segura et al. 2009). In Sierra Nevada, temperature is the main limiting factor of biological activity at the highest altitudes (oro- and crioro-mediterranean bioclimatic belts, see Figure in response R14), where vegetation growth is centered in the summer months (high temperature and water availability from the thaw). However, lower altitudes, and particularly in the southern and eastern parts of Sierra Nevada (see Figure in response R14), have drier and warmer conditions, and vegetation activity is mainly constrained by water availability during the summer. In these areas, during the winter months, precipitation is greater and temperature is still mild (12-15 °C, Rivas-Martínez 1997) which allows for vegetation growth (Alcaraz-Segura et al. 2009). In addition, such environmental conditions in the drylands have shaped plant species adaptations that enhance peak winter greenness, due to their fast response to scarce water inputs ( Cabello et al. 2012). This is the case of some of the meso- and thermo- mediterranean scrublands (e.g. Anthyllis terniflora scrublands) which are dominated by summer-deciduous or malacophilous plant species (e.g. Anthyllis cytisoides, A. terniflora) that develop their maximum foliage in

the mid-winter. Nevertheless, ecosystem functioning in these areas was considered as rare in the context of the whole reserve, presenting EFTs with maximum greenness in winter (Ba4, Aa4). Since these areas are at the lowest altitudes in the park, and are not affected by the snow, the data filtering will not affect their EVI max in the winter period.

Another interesting result for the winter EVI peaks that can also be related to the particular climatic conditions in Mediterranean areas, is the case of pine afforestations at the upper bioclimatic belts (mainly in the supramediterranean belt). In Sierra Nevada, the pine formations in these bioclimatic belts correspond mainly to Pinus sylvestris and Pinus pinaster plantations and occur in the North slope, where we have found the largest areas with winter EVI max. According to Aragonés et al. (2019), there is a generalised pattern in the warmest and driest Mediterranean areas (i.e. Iberian Peninsula), where growing seasons of pines begin in autumn and extend to the following spring, due to the mild winters. The dormant period for these formations occurs in summer, as a consequence of the prolonged water stress (Atzberger et al. 2013, Peñuelas et al. 2004, Verger et al. 2016). Aragonés et al. (2019) findings couple with the phenological patterns in the supra- and oromediterranean pine forest areas, showing maximum peaks of EVI in winter (Fig. 7). Hence, Mediterranean pine species differing in relation to the dates of phenological events of the northern hemisphere, where the typical growing season of the vegetation in the northern hemisphere, according to the NDVI phenological pattern, starts with the photosynthetic activity in spring, achieves its maximum at the beginning of summer, and ends in autumn, with a dormant period in winter.

C11. - * Line 161: "relative extension" - what do you mean, here? Share of area of EFTi within a defined area (moving window)?

R11. - Thank you for this comment, which can lead to misinterpretation. "Relative extension" means that, in order to calculate rarity, the abundance of each EFT (in terms of occupied surface) is relative to the most abundant. However, for a better understanding, we will rewrite this part and the phrase will be replaced by removing the word "relative": "EFT rarity was calculated as the extension of each EFT compared

to the most abundant EFT"

Once we have the rarity value of each EFT (using Equation 1) (line 164), we assign to each pixel in the EFT map such value according to its EFT class. Hence, the original spatial resolution of the EFT rarity map is the same as the resolution of the EFT map (230 m).

C12. - * Line 162: "compared to the most abundant EFT" – in a defined area / window?

R12. - Compared to the most abundant EFT in the study area. We will add this clarification to the sentence in the manuscript as follows: "EFT rarity was calculated for each year as the relative extension of each EFT compared to the most abundant EFT in the study area (Equation 1)" (line 164).

C13. - * Line 218: "altitudinal patterns"- What about topographical patterns (aspect, slope)?

R13. Thanks for this comment. We talk about data referring to altitudinal patterns as an example of data description. In the revised version, we will include patterns related to topography. However, we consider that further formal analyses in this regard are beyond the scope of a descriptive data paper.

C14. - * Line 219 ff.: I cannot find any map of those bioclimatic belts for the study area. Hence, I am not able to follow the description of results. Please add a figure.

R14. - We are very thankful for this comment, which will allow a better reading of the results. In the new manuscript, we will modify Figure 1 in the manuscript to include: the delimitation of the Biosphere Reserve and the distribution of the main ecosystems (Pérez-Luque et al. 2019) and thermotype bioclimatic belts (Molero-Mesa and Marfil 2015). We have added the new Figure 1 here, it is Figure 8 on this letter.

C15. - * Line 235: "maximum greenness in winter" – see comment above, how would you explain a greenness peak in wintertime?

R15. - Please, see response R10.

C16. - * Line 254: "interannual variability ranged from 1 to 17 different EFTs over the 18-year period" - what is the contribution of data uncertainty / data quality in this context, e.g. the missing QA-masking on one side and the very low EVI values on the other hand?

R16. - We consider that the data quality has no significant effect on the interannual variability of EFTs in the study area. First, although the inter-annual variability ranged from 1 to 17 different EFTs over the 18-year period, that maximum value of EFT changes occurred in only two pixels of the study area. More than 90% of the study area showed less than 10 EFTs over the 18-year period, and only 3% of the study area showed more than 12 EFTs. More than 75% of the study area showed a variability from 1 to 8 different EFTs (Table 4). Furthermore, variability was greater in intermediate biocli-matic belts, e.g. the mesomediterranean or supramediterranean, where pixels are not so influenced by the snow, but are more exposed to varying limiting conditions among years, summer droughts some years and winter cold some others. Whereas in the highest bioclimatic belts (e.g. oro- crioromediterranean), where the presence of snow and clouds is greater and more regular, so data quality would have a greater influence, the interannual variability was lower.

Second, regarding the very low EVI values (i.e. negative values), we had already transformed all negative values into zeros (but it was not sufficiently explained in the manuscript), to remove the potential remaining effect of snow, ice and water (please, see R1). Thus, we do not expect a high effect of filtering snow, ice or water on interan-nual variability of EFTs.

Finally, considering the small effect of filtering using the Quality Assessment bands on Ecosystem Functional Attributes, we believe that filtering would not affect the inter-annual variability. However, if the reviewer and editor still think that we should apply the QA filtering, we will filter out snow, ice and water as zeros and clouds as NANs.

C17. - * Line 359: "geospatial data Sierra Nevada Park" – Where do you show these data?

R17. - This is just the shapefile with the boundaries of Sierra Nevada. We will rename it and we will also include the ecosystem and bioclimatic belt maps (please, see R14).

C18. - * Line 366: "Sierra Nevada Biosphere Reserve (SE Spain)" – show in map!

R18. - Thank you for your suggestion, we will add a new figure showing this one, see R14.

C19. - * Figure 1: It would be helpful for the interpretation of the EFA and EFT data to have a map of vegetation types rather than a simple snapshot from the ISS without any information on content and scale.

R19. - We will do that, see R14.

C20. - * Figure 3: the mean EVI is NOT the "area under curve"! This would rather be the cumulative EVI.

R20. - That's right, we will rewrite this sentence as follows: " EFAs were: the annual mean or the cumulative EVI, an estimator of annual productivity (EVI_mean), the EVI seasonal coefficient of variation, i.e. the differences between the minimum and the maximum EVI values, a descriptor of seasonality (EVI_sSD), and the date of maximum EVI, an indicator of phenology (EVI_DMAX)".

REFERENCES

Alcaraz, D., Paruelo, J., & Cabello, J. (2006). Identification of current ecosystem functional types in the Iberian Peninsula. Global Ecology and Biogeography, 15(2), 200-212.

Alcaraz-Segura, D., Cabello, J., & Paruelo, J. (2009). Baseline characterization of major Iberian vegetation types based on the NDVI dynamics. Plant Ecology, 202(1), 13-29.

Anderson, C. B. (2018). Biodiversity monitoring, earth observations and the ecology of scale. Ecology letters, 21(10), 1572-1585.

Aragones, D., Rodriguez-Galiano, V. F., Caparros-Santiago, J. A., & Navarro-Cerrillo, R. M. (2019). Could land surface phenology be used to discriminate Mediterranean pine species?. International Journal of Applied Earth Observation and Geoinformation, 78, 281-294.

Atzberger, C., Klisch, A., Mattiuzzi, M., & Vuolo, F. (2014). Phenological metrics derived over the European continent from NDVI3g data and MODIS time series. Remote Sensing, 6(1), 257-284.

Cabello, J., Alcaraz-Segura, D., Ferrero, R., Castro, A. J., & Liras, E. (2012). The role of vegetation and lithology in the spatial and inter-annual response of EVI to climate in drylands of Southeastern Spain. Journal of Arid Environments, 79, 76-83.

de Beurs, K. M., & Henebry, G. M. (2010). Spatio-temporal statistical methods for modelling land surface phenology. In Phenological research (pp. 177-208). Springer, Dordrecht.

Didan, K. (2015a). MOD13Q1 MODIS/Terra Vegetation Indices 16-Day L3 Global 250m SIN Grid V006 [Data set]. NASA EOSDIS Land Processes DAAC. Accessed 2020-03-25 from https://doi.org/10.5067/MODIS/MOD13Q1.006

Didan, K., Munoz, A. B., Solano, R., & Huete, A. (2015b). MODIS vegetation index user's guide (MOD13 series). University of Arizona: Vegetation Index and Phenology Lab.

Henebry, G. M., & de Beurs, K. M. (2013). Remote sensing of land surface phenology: A prospectus. In Phenology: An integrative environmental science (pp. 385-411). Springer, Dordrecht.

Huete, A. R., Liu, H. Q., Batchily, K. V., & Van Leeuwen, W. J. D. A. (1997). A comparison of vegetation indices over a global set of TM images for EOS-MODIS. Remote

sensing of environment, 59(3), 440-451.

Huete, A., Didan, K., Miura, T., Rodriguez, E. P., Gao, X., & Ferreira, L. G. (2002). Overview of the radiometric and biophysical performance of the MODIS vegetation indices. Remote sensing of environment, 83(1-2), 195-213.

Huete, A., Justice, C., & Van Leeuwen, W. (1999). MODIS vegetation index (MOD13). Algorithm theoretical basis document, 3(213).

Ivits, E., Cherlet, M., Horion, S., & Fensholt, R. (2013). Global biogeographical pattern of ecosystem functional types derived from earth observation data. Remote Sensing, 5(7), 3305-3330.

Liu, H. Q., & Huete, A. (1995). A feedback based modification of the NDVI to minimize canopy background and atmospheric noise. IEEE transactions on Geoscience and Remote Sensing, 33(2), 457-465.

Lourenço, P., Alcaraz-Segura, D., Reyes-Díez, A., Requena-Mullor, J. M., & Cabello, J. (2018). Trends in vegetation greenness dynamics in protected areas across borders: what are the environmental controls?. International Journal of Remote Sensing, 39(14), 4699-4713.

Molero Mesa, J. & Marfil, J.M. 2015. The bioclimates of Sierra Nevada National Park. Int. J. Geobot. Research 5: 1-11.

Monteith, J. L. (1972). Solar radiation and productivity in tropical ecosystems. Journal of applied ecology, 9(3), 747-766.

Paruelo, J. M., & Lauenroth, W. K. (1995). Regional patterns of normalized difference vegetation index in North American shrublands and grasslands. Ecology, 76(6), 1888-1898.

Paruelo, J. M., Jobbágy, E. G., & Sala, O. E. (2001). Current distribution of ecosystem functional types in temperate South America. Ecosystems, 4(7), 683-698.

Paruelo, J. M., Texeira, M., Staiano, L., Mastrángelo, M., Amdan, L., & Gallego, F. (2016). An integrative index of Ecosystem Services provision based on remotely sensed data. Ecological indicators, 71, 145-154.

Peñuelas, J., Filella, I., Zhang, X., Llorens, L., Ogaya, R., Lloret, F., ... & Terradas, J. (2004). Complex spatiotemporal phenological shifts as a response to rainfall changes. New Phytologist, 161(3), 837-846.

Pérez-Luque, A.J.; Bonet-García, F.J.; Zamora Rodríguez, R (2019). Map of Ecosystems Types in Sierra Nevada mountain (southern Spain). PANGAEA, https://doi.org/10.1594/PANGAEA.910176

Pettorelli, N., Vik, J. O., Mysterud, A., Gaillard, J. M., Tucker, C. J., & Stenseth, N. C. (2005). Using the satellite-derived NDVI to assess ecological responses to environmental change. Trends in ecology & evolution, 20(9), 503-510.

Requena-Mullor, J. M., Reyes, A., Escribano, P., & Cabello, J. (2018). Assessment of ecosystem functioning from space: Advancements in the Habitats Directive implementation. Ecological Indicators, 89, 893-902.

Townshend, J. R., Goff, T. E., & Tucker, C. J. (1985). Multitemporal dimensionality of images of normalized difference vegetation index at continental scales. IEEE Transactions on Geoscience and Remote sensing, (6), 888-895.

Verger, A., Filella, I., Baret, F., & Peñuelas, J. (2016). Vegetation baseline phenology from kilometric global LAI satellite products. Remote sensing of environment, 178, 1-14.

Virginia, R. A., Wall, D. H., & Levin, S. A. (2001). Principles of ecosystem function. Encyclopedia of biodiversity, 2, 345-352.

Please also note the supplement to this comment:
https://www.earth-syst-sci-data-discuss.net/essd-2019-198/essd-2019-198-AC1-

supplement.pdf

Stack of 16 daily
images

Pixel by pixel analysis

No — n good
QA Pixels
≥ 2 — Yes

No — n avail.
Pixels ≥ 1 — Yes

Compute Fill
value using
historic average

Select best
pixel using
MVC

Select best
pixel using CV-
MVC

Image integrated
pixel by pixel

Resulting
composite tile

**Fig. 1.** MODIS compositing algorithm data flow (from Didan et al. 2015b).

[Figure]

Figure 2. Comparisons of EFAs with MODIS MOD13Q1 EVI data filtered by quality (layer product "QA summary": bands 2 (snow/ice) and 3 (clouds)) and with pre-processing data of the product and negative values (generally corresponding to snow, ice or water) converted into zeros. Left panels (a, c, e) show maps of simple sliding window (3x3 pixels) correlation for EVI_mean and EVI_SD (continuous variables) and changes for EVI_DMAX (discrete variable). Right panels (b, d, f) show the histograms associated with the left panel maps, i.e. how many pixels have a low or high correlation (b, d) and for the case of EVI_DMAX how many pixels have changed or not (f).

**Fig. 2.**

[Figure]

[Figure]

**Fig. 3.** Scatter plots between the non-filtered (X-axes) and filtered (Y-axes) EFAs: a) EVI_mean MOD13Q1.006 and b) EVI_SD MOD13Q1.006, both of the average year.

[Figure]

**Fig. 4.** Effect of Scan Line Corrector fault on Landsat7 imagery in Sierra Nevada and data gaps due to clouds (in green and white). Landsat-7 image courtesy of the U.S. Geological Survey.

Spatiotemporal scales of biodiversity
measurements from Earth observations

Figure 5. Anderson (2018) : *"Log–log plot of spatial and temporal and grain sizes for 44 current and historic satellite Earth observation (EO) sensors, coloured by biodiversity pattern type. Several sensors have been used to measure multiple biodiversity patterns, and the most cited or most novel were selected in these cases".*

**Fig. 5.**

[Figure]

**Fig. 6.** Eigenvectors of the first two components of a PCA performed on the annual curve of
EVI values in Sierra Nevada (Axis x: months; axis y: eigenvectors values).

[Figure]

Figure 7. From Aragonés et al. (2019). "*NDVI values in the average year. This shows the average values of the sum of the trend and seasonal components for each species, obtained using the BFAST algorithm. Three months of data are shown before and after, as the growing seasons can span the calendar year.*" This study was carried out in the Iberian Peninsula, including samples from Sierra Nevada.

**Fig. 7.**

[Figure]

Figure 1. Study area: Sierra Nevada Biosphere Reserve. a) Location in the context of the Iberian Peninsula; b) remote view of Sierra Nevada mountain region (image from the International Space Station took in December 2014; courtesy of "Earth Science and Remote Sensing Unit, 615 NASA Johnson Space Center"); c) delimitation of the Biosphere Reserve and the distribution of the main ecosystems (Pérez-Luque et al. 2019) and thermotype bioclimatic belts (Molero-Mesa and Marfil 2015).

**Fig. 8.** NEW FIGURE 1 OF MANUSCRIPT

| Scores | | | | | | | | | | | | | |
|---|---|---|---|---|---|---|---|---|---|---|---|---|---|
| PCA axis | %[a] | Jan | Feb | Mar | Apr | May | Jun | Jul | Aug | Sep | Oct | Nov | Dec |
| 1 | 87.3 | 0.334 | 0.328 | 0.333 | 0.318 | 0.293 | 0.246 | 0.236 | 0.239 | 0.242 | 0.251 | 0.287 | 0.325 |
| 2 | 96.5 | 0.329 | 0.365 | 0.326 | 0.109 | -0.244 | -0.454 | -0.380 | -0.301 | -0.252 | -0.154 | -0.007 | 0.229 |

[a] *Cumulated variance*

**Fig. 9.** Table 1. Eigenvectors and cumulative variance explained by the first two components of a principal component analysis (PCA) performed on the annual curve of EVI values in Sierra Nevada.

|  | PC1 | PC2 |
|---|---|---|
| **EVI_Mean** | **0.94** | -0.01 |
| **EVI_SD** | -0.14 | **-0.75** |
| **DMAX_Sine** | -0.10 | **0.67** |
| **DMAX_Cosine** | 0.017 | **-0.61** |

**Fig. 10.** Table 2. Correlation values between PCA axis 1 and 2 and ecosystem functional attributes.

|  | EVI_Mean | EVI_SD |
|---|---|---|
| EVI_Mean | 1 |  |
| EVI_SD | -0.14 | 1 |
| EVI_DMAX | 0.10 | -0.05 |

**Fig. 11.** Table 3. Pearson correlation values between metrics.

| Interannual variability | n pixels | % study area surface | Cumulated % of study area surface |
|---|---|---|---|
| 1 | 49 | 0.12 | 0.12 |
| 2 | 301 | 0.74 | 0.86 |
| 3 | 2582 | 6.32 | 7.17 |
| 4 | 4436 | 10.85 | 18.02 |
| 5 | 6010 | 14.70 | 32.73 |
| 6 | 6605 | 16.16 | 48.88 |
| 7 | 6064 | 14.83 | 63.72 |
| 8 | 5144 | 12.58 | 76.30 |
| 9 | 3875 | 9.48 | 85.78 |
| 10 | 2707 | 6.62 | 92.40 |
| 11 | 1678 | 4.11 | 96.51 |
| 12 | 871 | 2.13 | 98.64 |
| 13 | 372 | 0.91 | 99.55 |
| 14 | 130 | 0.32 | 99.87 |
| 15 | 40 | 0.10 | 99.97 |
| 16 | 12 | 0.03 | 99.99 |
| 17 | 2 | 0.01 | 100.00 |

**Fig. 12.** Table 4. Number of pixels and percentage of the study area that experienced different levels of inter-annual variability in EFTs (number of EFTs that were observed over the period in each pixel).

---

## Referee Comment (RC3) · Anonymous Referee #1 · 15 Apr 2020

Dear authors,
thank you very much for your response to my comments.
I very much appreciate your detailed investigations and explanations on the issues that I raised. You fully considered my points and I do not have any further requests for revisions.

There is only one very minor, technical point:
Fig. 1B in your new figure is missing a scale and the orientation differs from the new

map that you included in 1C. Could you please fix this?

Kind regards!

---

## Author Comment (AC2) · 25 Apr 2020

Dear Reviewer,

Many thanks for your correspondence regarding our data description paper entitled "A remote sensing-based dataset to characterize the ecosystem functioning and functional diversity of a Biosphere Reserve: Sierra Nevada (SE Spain)". We thank you for all your constructive comments, which provided valuable insights to improve the conceptual and methodological robustness of our data and our manuscript. We are now

very pleased to send you the response to your comments and suggestions.

In our response below, please find our point-by-point responses (indicated with "R") presenting, in detail, how we have addressed the Reviewer comments ("C"). In the .pdf document attached, the Reviewer comments are reproduced in bold italic font and our responses are indicated in plain text, in addition, tables and figures are embedded in the main document. We numbered each comment and reply for ease of reference and indicated changes that will be made in the manuscript, which will be submitted after the open discussion.

Once again, we thank you for your time, constructive comments and suggestions. We hope to meet the expectations with this response, and that the Reviewer considers our data description manuscript suitable to be published in Earth System Science Data.

Sincerely,

The authors

C1. - * Are the data and methods presented new?  -An interesting approach is presented for inter-annual heterogeneity; it is left open why for assessing the spatial variability a certain kernel size had been chosen

R1. - Thank you for your positive comment. Regarding kernel size, we chose a 4x4-pixel kernel as a balance between spatial resolution and saturation of the EFT richness variable. That is, using kernels of 2x2 and 3x3 pixels resulted in a high proportion of kernels that reached the highest possible richness value (4 and 9 EFT classes per kernel, respectively), so the EFT richness variable was highly saturated. Using kernels of 5x5 or greater number of pixels never saturated the maximum number of pixels in a kernel but resulted in too coarse outputs (grain size greater or equal to 5x5 pixeles). The 4x4 kernel offered the finest spatial resolution of the EFT richness map and was never saturated. In other words, the maximum EFT richness within a 4x4-pixel kernel that we registered was 13, but the potential maximum number could have been 4x4=16

(Fig. 1).

We will add in the text the justification for this choice, section 2.5, as follows: "We chose a 4x4-pixel window since it offered the finest spatial resolution without saturating the number of EFT classes per kernel (i.e. smaller sizes result in a high proportion of kernels with the maximum number of classes)". We can also add an appendix with the Fig. 1 included in this response letter.

Any richness measurement exercise depends on spatial scale (i.e., both grain and extent) of assessment (Arponen et al., 2012). Regarding grain, when using species distributions to identify hotspots, the actual values of species richness found in each cell will increase with grain from a dataset built at 1x1 km to a dataset built at 10x10 km. However the regional spatial patterns of species richness will not vary widely (Rahbek 2005). In our analysis, the maximum number of EFTs found in a kernel could also vary depending on the kernel size, as stated above. If we used smaller kernel sizes, we would find lower and saturated EFT richness values. By contrast, with a larger kernel size (e.g. 5x5), the observed patterns would be too coarse.

C2. - * Is there any potential of the data being useful in the future? -In principle yes, however, there are details missing, see next

R2. - Thank you very much for the comment, as shown by numerous works cited in the manuscript (section 4), ecological research based on spectral vegetation indices plays an important role in biodiversity conservation (Cabello et al., 2012; Pettorelli, 2016, 2018) and management (Pelkey et al., 2003; Cabello et al., 2016) and for the study of biodiversity and ecosystems responses to environmental changes (Pérez-Luque et al., 2015; Alcaraz-Segura et al., 2017). In particular, our dataset provides valuable information to the scientific community as an example of a novel and straightforward characterization of functional diversity at ecosystem level developed for an entire pro-tected area. This approach can be exported to any protected area to help incorporate the ecosystem functional dimension into conservation practice. Since Sierra Nevada

Biosphere Reserve is a Long-Term Ecological Research site established 10 years ago (Zamora et al., 2016, 2017), our dataset compliments many others on biodiversity, climate, ecosystem services, hydrology, land-use changes and management practices in the area. This further increases the value of the data to the scientific community, since it makes now possible to explore the relationships between previous biodiversity and environmental data with the ecosystem functional data that we provide (section 4 in the manuscript).

C3a. - * Are methods and materials described in sufficient detail? - No. Why is the kernel size 4x4?

R3a. - Thank you very much for raising this question. Please, see R1, where we justify the choice of that kernel size. In addition, we will add in the text the justification for this choice, section 2.5, as we indicated in R1.

C3b.- * How have borderline pixels be processed with the kernel? (kernel processed raster layer have same extension)

R3b. - Thank you for this warning. We will specify this process in the manuscript in section 2.5 as follows: "Note that since we only classified MODIS pixels within the protected area, the 4x4-pixel sliding windows along the borderline of the protected area that contained pixels outside it (classified as NoData) could probably contain a lower EFT richness value in the dataset than in reality."

In addition, if the editor and referees consider the next paragraph useful, we can explain that to avoid pixels outside the protected area with NoData values being considered as a distinct class when calculating EFT richness, we processed as follows: 1) first, we built a 0-1 mask by rasterizing the vector boundaries of the study area to the same pixel size and projection of the MOD13Q1 product; 2) second, we used the same kernel used for EFT richness to obtain those kernels with pixels along the border where NoData could artificially increase richness by 1; 3) then, we subtracted this last output to the original non-corrected EFT richness image to correct the artificial increase of

richness due to NoData values outside the borders.

C4.- * How variable are the quartile boundaries (could you name a standard deviation?)

R4. - Thank you very much for the suggestion, we believe that adding this information to the manuscript will add value to the data. To know how variable the quartiles were, we will show the quartiles of each year, their interannual mean, their interannual standard deviation, and their interannual coefficient of variation (Table 1/Fig.4 in this document). The variability among years or Coefficient of Variation (CV) was around 5% for EVI_mean quartiles and lower than 11% for EVI_SD quartiles, increasing in the uppest quartiles (Table 1).

Having such interannual variability in the quartiles shows the influence that climate fluctuations (e.g. dry or wet years) have on vegetation greenness. As we will further explain in the manuscript, we developed a fixed-classification approach with fixed limits between classes for the entire period so that our EFT classification was capable of capturing such inter-annual changes. Adapting the limits between classes to each year would not make possible to compare the classification across years.

C5.- * Are any references/citations to other data sets or articles missing or inappropriate? -reference/URL to the database REDIAM is missing, also, which particular datasets have been employed from it; by what data got the MODIS data clipped/masked?

R5. - Thank you for pointing out the missing reference. The MODIS data were clipped by the shapefile with the boundaries of Sierra Nevada protected area obtained from REDIAM, the public repository of environmental information of the Andalusian government. The REDIAM URL will be added to the manuscript: http://www.juntadeandalucia.es/medioambiente/RENPA.

C6.- * Is the article itself appropriate to support the publication of a data set? - yes with respect to gain an understanding of the data. The article does not provide necessary

information to re-use the data: the legend for EFTs is part of Fig 2; the values of the EFTs do not correspond to the values in the TIFs (there they are 1-64 encoded)

R6. - Thank you very much for pointing out this confusing issue. The legend in Figure 2d of the manuscript has numerical values (from 1 to 64) and their corresponding EFT codes (from 1=Aa1 to 64=Dd4) (Fig. 2). The .TIFs files only include the numerical coding from 1 to 64 since it is not possible to store alphanumeric (string or character) information in .TIF. However, the corresponding alphanumeric codes can be easily consulted in the legend. We will clearly explain this in the manuscript (section 2.4) and include it in the corresponding metadata files: " The EFT alphanumeric code (Aa1 to Dd4) corresponding to the numeric code (1 to 64) in the .TIF files is contained in the legend of Figure 2d".

C7.- * Check the data quality: Is the data set accessible via the given identifier? -yes Is the data set complete? -yes Are error estimates and sources of errors given (and discussed in the article)? - well, not error but there is no reference to variability eg the means of internal quartiles given

R7. - Please, see the responses R4, R29a and R29c, where we explained how we handled the variability in the quartiles, which will be included in the new version of the manuscript.

C8. - * Are the accuracy, calibration, processing, etc. state of the art? - The article employs community-"standard" pre-processed data; however, it does not provide accuracy information of intermediate processing steps. Also, the derivation of spatial heterogeneity, the chosen size of the kernel and how this affects the results is not discussed

R8. - Accuray information of the intermediate steps of the process are documented in the R4, R29a, R29c, R1 and R3b, in addition, the effect of kernel size on our results will be added and discussed in the new version of the manuscript.
C9.- * Are common standards used for comparison? - the resulting data are not compared Is the data set significant – unique, useful, and complete? -The data set is useful

R9. - Thank you for your encouraging comments.

C10.- * Consider article and data set: Are there any inconsistencies within these, implausible assertions or data, or noticeable problems which would suggest the data are erroneous (or worse). - using a kernel to derive values I would have expected that the resulting layer is smaller in size than the input layer, unless some "mirroring" is done to extend the input layer in size. The article does not provide any information on how this was handled

R10. - The output layer has the same size as the input layer because the kernel assigns to each pixel the value of EFT richness by counting how many different EFTs there are in the surrounding 4x4 pixels, therefore the output resolution and layer size is the same. To provide information on how this was handled, we will add a sentence explaining it in section 2.5, in addition to the kernel size justification (R1), as follows: "EFT richness was calculated for each year by counting the number of different EFTs in a $4\times4$-pixel moving window around each pixel (top-left center pixel of the 4x4 Kernel) (modified from Alcaraz-Segura et al., 2013). Each MODIS pixel received a richness value derived from counting how many different EFTs there were in the surrounding 4x4 pixels. We chose a 4x4-pixel window since it offered the finest spatial resolution without saturating the number of EFT classes per kernel (i.e. smaller sizes result in a high proportion of kernels with the maximum number of classes). This is the reason why all images in the dataset have the same number of columns and rows".

Also, we have explained the handling of the kernel in the R1, R3a, R3b.

C11.- * If possible, apply tests (e.g. statistics). - looking up the TIFs with standard GIS software(QGis) did not reveal any problems. The histograms of values seem ok, although because of missing legend they could not be really interpreted

[Figure]

R11.- Please, see R6 for explanation of .TIFs values and legend.

C12.- * Is the data set itself of high quality? Check the presentation quality: Is the data set usable in its current format and size? -yes, the GeoTIFF is a well accepted and documented file format Are the formal metadata appropriate? - No, I am unable to discover any formal metadata. The GeoTIFF come with some metadata in their header, but do require specialized software for extraction, eg. of the bounding box or employed projection. additional TFW file would be readable with common editors. Additional formal metadata is missing.

R12. - We will made a Data Management Plan with the formal metadata of our dataset as in this example: https://dmptool.org/plans/8278/export.pdf As the reviewer points out, our .TIFs files already contain this metadata: raster information (columns and rows, number of bands, cell size, uncompressed size, format, source type, pixel type, pixel depth, NoData value, pyramids, compression, status), extension (top, left, right, bottom), spatial reference (angular unit, datum) and statistics (build parameters, min, max, mean, std dev.). Thus, considering the available metadata and the very time-consuming effort that represents reprocessing all data with an additional .tfw file along with the metadata contained in each archive .TIF, we consider that a document on metadata such as the Data Plan Management could give the necessary information in terms of metadata. However, if the reviewer and editor still think that we should provide one .TFW file per .TIF, we can reprocess all the data to make it.

C13.- * Check the publication: Is the length of the article appropriate? - given, that it is a data publication, the article dwells much on discussion of the application/biodiversity/structure but is much shorter when it comes to describing data and methodology

R13.- Thank you for your comment. As already stated in other responses, in the new version of the manuscript, which will be submitted after the open discussion, we will expand the description of the data and methodology.

[Figure]

C14.- * Is the overall structure of the article well structured and clear? -yes Is the language consistent and precise? -there are a few language errors but the article is language wise in good shape

R14. - We are very thankful for the Reviewer's encouraging remarks! To improve remaining language errors, we will thoroughly review English grammar and spelling.

C15.- * Are mathematical formulae, symbols, abbreviations, and units correctly defined and used? - Eq.3 uses X any Y without explicit definition; this equation does not provide additional information content

R15. - Equation 3 refers to the Jaccard index: J(X,Y) = |X âĹ' Y| / |X âĹł Y| , where the Jaccard index for two data sets (X = set 1; Y =set 2) is equal to the size of the intersection divided by the size of the union of the data sets. In the new manuscript, we will give the explicit definition of X and Y in the same way as in this response.

C16.- * Are figures and tables correct and of high quality? Quality is mostly acceptable, in Fig.2, part 3 the legend is hardly readable

R16.- Thanks for the advice, we will increase the quality of the legend in Figure 2 of the manuscript.

C17.- * Finally: By reading the article and downloading the data set, would you be able to understand and (re-)use the data set in the future? -No, eg. the EFT type as encoded in the TIFs cannot be interpreted

R17.- Please, see R6.

C18.- * Uniqueness: It should not be possible to replicate the experiment or observation on a routine basis. - all resulting data can be reproduced as the primary source is generally available. However, the derivation needs expertise with GIS/remote sensing software, and a target audience of ecologists is usually easier reached with data products which are deemed useful for such clientele

R18. - Our goal providing this dataset is to give the scientific community an example of how to derive valuable information of the functional diversity at ecosystem level for an entire protected area. We provide this dataset for the LTER site of Sierra Nevada Biosphere Reserve so that other researchers and managers can use it without the need for remote sensing expertise. However, we provide all the information and data sources to be reproducible by those experts who wish to reproduce it in this or any other area of the world.

C19.- * The introduced methods are not trivial nor obvious, however, would benefit from a discussion why certain approaches had been taken (kernel size eg.)

R19. - Please, see R1, R3b.

C20.- * The data seem complete. All derived data sets are provided (annual data), also the summary data. In theory one could re-calculate all results (if eg. interval boundaries were to be now known, EVI_max).

R20. - The intervals of months to define each season and therefore EVI_max were as follows: January to March = 4 - Winter. April to June = 1 - Spring. July to September = 2 - Summer. October to December = 3 - Autumn.

This information is important to appear in the manuscript to ensure its reproducibility, therefore it will be added in the next version.

C21.- * I would request information on hardware and software used to derive products (algorithmic deviations)

R21. - Most of processing was carried out in the Google Earth Engine (GEE) platform. GEE combines a multi-petabyte catalog of satellite imagery and geospatial datasets with planetary-scale analysis capabilities. We used the main Javascript programming interface to build the algorithms and requests to GEE servers. More information in https://earthengine.google.com/faq/ and https://developers.google.com/earth-engine/. Only inter-annual variability was processed with IDL software (short for Interactive

Data Language). IDL is commonly used for interactive processing of large amounts of data, including image processing. The syntax includes many constructs from Fortran and some from C. More information in https://www.harrisgeospatial.com/Software-Technology/IDL..

C22.- * Also, to reproduce the data information on masking/clipping the covered regions is necessary but absent. (which dataset, which method)

R22. - The data were clipped with the shapefile of the Sierra Nevada Biosphere Reserve boundaries, whose layer is available at REDIAM, (see R5). The method applied to extract the data was clipping MODIS data with the shapefile that delimited the Biosphere Reserve.

Technical details:

C23.- * line24: imagery do not provide a continuous characterization as reflectance is integrated per pixel

R23. - We agree with the reviewer, imagery does not provide a continuous characterization as reflectance is integrated per pixel, however this sentence refers to spatially explicit information (i.e. covering the whole territory). Therefore, as the sentence can be confusing, we will change the term by "spatially explicit" and we will rewrite the sentence as follows: "Nowadays, the use of satellite imagery provides useful methods to produce a spatially explicit characterization of ecosystem functioning and processes at regional scales".

C24.- * line 26: from 2001 to 2018

R24. - We will change "since" for "from", thank you for correcting this mistake.

C25.- * line 79 not the EFT approach has exp. grown but the application of EFT approaches

R25. - Thank you for your suggestion. We will change the sentence to "Since the

concept appeared in 2001 (Paruelo et al., 2001), the EFT approach (or equivalent approaches) applications has exponentially grown to characterize functional hetero-geneity from local to global scales (...)."

C26.- * line 137 EFT seasonal curve: the term has not been introduced properly; I presume it refers to the 23 measurements taken per year, please clarify

R26.- Yes, the seasonal curve refers to the 23 measurements per year. We will change the sentence to as follows: "These attributes were calculated from the EVI seasonal curve or annual dynamics (i.e. 23 measures per year)".

C27.- * line 146: one cannot understand the present derivation as the methodology is referred to another article; worse, the authors write of a "similar" approach without making clear how/where they differ

R27. - We note that it is similar to other articles and explain next what it is. The calculation of EFTs does not differ methodologically from the article mentioned, but methodological novelties from the concept are explained in the following sections (2.5, 2.6).

C28.- * line 147 EVI_DMAX: unclear, whether you chose the intervals according to the definition of the seasons or you derived them and they turned out to coincide with the seasons; please clarify

R28. - We chose the intervals of EVI_DMAX according to the definition of the seasons. Please, see R20. To clarify it in the manuscript, we will change the sentence as follows: "For EVI_DMAX, the four intervals according to the definition of the four seasons of the year: January to March = Winter, April to June= Spring, July to September = Summer, October to December = Autumn".

C29a.- * line 149-150: the derivation of quartile borders was understandable only after consulting the reference. How stable are the boundaries, that is, provide a standard deviation for each mean

R29a.- We will better explain how we used quartiles to define the limits between classes to make this manuscript self-standing and independent from our previous works. Regarding the stability of quartile boundaries across years, please, see our response R4. Table 1 in this letter indicates the quartile value for each year and the interannual mean that we used to set the limits between classes. In addition, it also contains the inter-annual standard deviation and coefficient of variation as indicators of the interannual variability associated with each mean (please, see R4 and R29c).

C29b.- Table 1: values cannot be reproduced nor checked, e.g. EVI_Mean_2001_C006_MOD13Q1_Pixel232.tif shows values between 11.5-4471.9 (QGis), table 1 reports 75% values are less than 0.241 EVI_mean:

R29b.- We thank the Reviewer for this useful comment so we can avoid misinterpretations from the readership. As the Reviewer points out, the .TIFs of EVI_Mean and EVI_SD files have values potentially ranging from 0 to 10,000, as indicated in line 131 of the manuscript as follows: "Values of EVI*10,000 are given as real numbers between 0 and 10,000". This is because the original EVI data ranged between those values to occupy less disk space. However, in the quartile table EVI_Mean and EVI_SD values were divided by 10,000, and therefore potentially ranging from 0 to 1.

We will include in the metadata and in the data management plan that in the EVI_Mean and EVI_SD .TIF files, values are multiplied by 10,000. We will also add the following information in the table heading (line 646): "Table 1. EFAs range used for identification of EFTs in Sierra Nevada. For EVI_DMAX, the four intervals agreed with the four seasons of the year. For EVI_mean and EVI_sSD, we extracted the first, second, and third quartiles for each year and then calculated the inter-annual mean of each quartile (their average over the 18-year period). The values of both EVI_mean and EVI_sSD are multiplied by 10,000 in the .TIF files to save disk space".

C29c. problem with "sealed" class boundaries: derivation relies on mean of a 18y period. If say, you want to show the time series of 2001-2020, would you need to do

the derivation of the boundaries or "extrapolate" from 2018?

R29c.- We developed a fixed-classification approach with "sealed" or fixed limits between classes for the entire period so that our EFT classification could detect inter-annual changes. Adapting the limits between classes to each year would not make it possible to compare the classification across years. For example, if there is a macro fire in 2020 over that burns the entire protected area, our use of fixed limits between classes will allow us to detect changes in EFTs in 2020 due to fire (most pixels would be classified as low productivity "A__ class"). However, if the limits between classes were adapted to each year, we would not detect in 2020 the effect of fire.

We determined the minimum number of years that are needed to reach stability in the quartile boundaries among classes. For each quartile, we plotted the maximum inter-annual coefficient of variation (Y axis) among the n consecutive years considered, with n ranging from n= 2 years to n=18 years against the number of years considered (X axis) (i.e. maximum value of the coefficient of variation among all possible combinations of two consecutive years, three consecutive years, four, five, etc. throughout the 2001-2018 period (Fig. 3). The three EVI_Mean quartiles tend to stabilize around an interannual coefficient of variation of 5%, which requires around 14 years of study period. The three EVI_SD quartiles tend to stabilize around an interannual coefficient of variation of 10%, which requires around 17 years of study period. Hence, the 18-year study period provided in this dataset would be enough to serve as a reference situation for this protected area. Thus, using the referee example, it would not be necessary to derive the quartiles boundaries again for the year 2020, since our 18-year study period is representative enough to extrapolate quartiles to the new year. We will include this analysis (including Fig. 3 in an appendix) and the referee example in the new version.

C30. - * Table 1, EVI_Max: values of 1-4 do not correspond to values found in TIFs (1-12)

R30. - The values of the ecosystem functional attributes appear with their original

values, in the case of EVI_max they are the months, i.e. as EVI mean and EVI SD are not grouped in 4, EVI mmax is not either. The values from 1 to 4 appear once we make the classification in groups to build the EFTs, but not in the EFAs map. We believe that providing the peak time with all months rather than the peak season (which is provided in the EFT map) is valuable, as it gives us greater yearly detail of the month of the phenology.

C31.- * line 159: justification for a 4x4 kernel? Why not 3x3 or 5x5? Could the kernel be dependend on the question being asked? How have borderline pixels be processed/why eg share richness and inter-annual mode the same borders?

R31. - Please see R1 and R3b.

C32.- * line 359: database is maintained

R32. - Thanks for the correction, we will change it in the manuscript.

C33.- * line 360: please include a reference/URL to the database REDIAM, also, indicate which datasets of REDIAM have been included in your work

R33.   - The data obtained from REDIAM was the shapefile with the boundaries of Sierra Nevada, which URL will be added to the manuscript: http://www.juntadeandalucia.es/medioambiente/RENPA.

C34.- * Fig 2.1; https://lpdaac.usgs.gov/products/mod13q1v006/ states 250m GSD, not 230m.

R34.- We strongly agree, but the 250m measure refers to the nickname of the dataset, not to the actual spatial resolution of the MOD13Q1 pixel, which is 231.65635826395828 m/pixel at the equator. We will explain this in the text and metadata.

C35.- * Fig 2.2: the mean is not the area under the curve, but the area normalized by the range; there is no curve at all but 23 discrete values/year

[Figure]

R35.- That's right, thank you, this was also pointed out by Reviewer 1. We will rewrite this sentence as follows: "EFAs were: the annual mean or the cumulative EVI, an estimator of annual productivity (EVI_mean), the EVI seasonal coefficient of variation, i.e. the differences between the minimum and the maximum EVI values, a descriptor of seasonality (EVI_sSD), and the date of maximum EVI, an indicator of phenology (EVI_DMAX)".

C36.- * Fig 2.4: the legend is crucial for reusing data but is not provided as individial data (eg. numerical values corresponding to a class, or pseudo color code for GoogleEarth); at present, the TIF files for eg EFTs show values between 1-64; how to map to your classes?

R36. - Please see R6.

REFERENCES

Alcaraz-Segura, D., Lomba, A., Sousa-Silva, R., Nieto-Lugilde, D., Alves, P., Georges, D., ... & Honrado, J. P. (2017). Potential of satellite-derived ecosystem functional attributes to anticipate species range shifts. International journal of applied earth observation and geoinformation, 57, 86-92.

Alcaraz-Segura, D., Paruelo, J. M., Epstein, H. E., & Cabello, J. (2013). Environmental and human controls of ecosystem functional diversity in temperate South America. Remote Sensing, 5(1), 127-154.

Arponen, A., Lehtomäki, J., Leppänen, J., Tomppo, E., & Moilanen, A. (2012). Effects of connectivity and spatial resolution of analyses on conservation prioritization across large extents. Conservation Biology, 26(2), 294-304.

Cabello, J., Alcaraz-Segura, D., Reyes, A., Lourenço, P., Requena, J. M., Bonache, J., ... & Serrada, J. (2016). Sistema para el seguimiento del funcionamiento de ecosistemas en la Red de Parques Nacionales de España mediante teledetección. Revista de Teledetección, 46, 119-131.

Cabello, J., Fernández, N., Alcaraz-Segura, D., Oyonarte, C., Pineiro, G., Altesor, A., ... & Paruelo, J. M. (2012). The ecosystem functioning dimension in conservation: insights from remote sensing. Biodiversity and Conservation, 21(13), 3287-3305.

Paruelo, J. M., Jobbágy, E. G., & Sala, O. E. (2001). Current distribution of ecosystem functional types in temperate South America. Ecosystems, 4(7), 683-698.

Pelkey, N. W., Stoner, C. J., & Caro, T. M. (2003). Assessing habitat protection regimes in Tanzania using AVHRR NDVI composites: comparisons at different spatial and temporal scales. International Journal of Remote Sensing, 24(12), 2533-2558.

Pérez-Luque, A. J., Pérez-Pérez, R., Bonet-García, F. J., & Magana, P. J. (2015). An ontological system based on MODIS images to assess ecosystem functioning of Natura 2000 habitats: A case study for Quercus pyrenaica forests. International Journal of Applied Earth Observation and Geoinformation, 37, 142-151.

Pettorelli, N., Schulte to Bühne, H., Tulloch, A., Dubois, G., MacinnisǎĂŘNg, C., Queirós, A. M., ... & Sonnenschein, R. (2018). Satellite remote sensing of ecosystem functions: opportunities, challenges and way forward. Remote Sensing in Ecology and Conservation, 4(2), 71-93.

Pettorelli, N., Wegmann, M., Skidmore, A., Mücher, S., Dawson, T. P., Fernandez, M., ... & Jongman, R. H. (2016). Framing the concept of satellite remote sensing essential biodiversity variables: challenges and future directions. Remote Sensing in Ecology and Conservation, 2(3), 122-131.

Rahbek, C. (2005). The role of spatial scale and the perception of largeǎĂŘscale speciesǎĂŘrichness patterns. Ecology letters, 8(2), 224-239.

Zamora Rodríguez, R. J., Pérez Luque, A. J., Bonet, F. J., Barea-Azcón, J. M., & Aspizua, R. (2016). Global change impacts in Sierra Nevada: challenges for conservation. Consejería de Medio Ambiente y Ordenación del Territorio. Junta de Andalucía, 208 pp.

[Figure]

Zamora, R., Pérez-Luque, A. J., Bonet, F. J., Barea-Azcón, J. M., Aspizua, R., Sánchez-Gutiérrez, F. J., ... & Henares-Civantos, I. (2017). Global change impact in the Sierra Nevada long-term ecological research site (Southern Spain). Bulletin of the Ecological Society of America, 98(2), 157-164.

Please also note the supplement to this comment:
https://www.earth-syst-sci-data-discuss.net/essd-2019-198/essd-2019-198-AC2-supplement.pdf

─────────────────────────

**EFT Richness
2x2 pixel kernel**

**EFT Richness
3x3 pixel kernel**

**EFT Richness
4x4 pixel kernel**

**Fig. 1.** EFT Richness for 2x2, 3x3 and 4x4-pixel kernel sizes. A 4x4-pixel kernel was chosen since it offered the finest spatial resolution that did not saturate the number of EFT classes per kernel.
**A-D: Productivity (increasing)**
**a-d: Seasonality (decreasing)**
**1-4: Phenology (Sp-Sm-Au-Wi)**

| | | | |
|---|---|---|---|
| 1) Aa1 | 17) Ba1 | 33) Ca1 | 49) Da1 |
| 2) Aa2 | 18) Ba2 | 34) Ca2 | 50) Da2 |
| 3) Aa3 | 19) Ba3 | 35) Ca3 | 51) Da3 |
| 4) Aa4 | 20) Ba4 | 36) Ca4 | 52) Da4 |
| 5) Ab1 | 21) Bb1 | 37) Cb1 | 53) Db1 |
| 6) Ab2 | 22) Bb2 | 38) Cb2 | 54) Db2 |
| 7) Ab3 | 23) Bb3 | 39) Cb3 | 55) Db3 |
| 8) Ab4 | 24) Bb4 | 40) Cb4 | 56) Db4 |
| 9) Ac1 | 25) Bc1 | 41) Cc1 | 57) Dc1 |
| 10) Ac2 | 26) Bc2 | 42) Cc2 | 58) Dc2 |
| 11) Ac3 | 27) Bc3 | 43) Cc3 | 59) Dc3 |
| 12) Ac4 | 28) Bc4 | 44) Cc4 | 60) Dc4 |
| 13) Ad1 | 29) Bd1 | 45) Cd1 | 61) Dd1 |
| 14) Ad2 | 30) Bd2 | 46) Cd2 | 62) Dd2 |
| 15) Ad3 | 31) Bd3 | 47) Cd3 | 63) Dd3 |
| 16) Ad4 | 32) Bd4 | 48) Cd4 | 64) Dd4 |

**Fig. 2.** EFT legend with numerical values (from 1 to 64) and their corresponding EFT codes (from 1=Aa1 to 64=Dd4).

Interactive
comment

[Figure]

Figure 3. Stabilization of the interannual coefficient of variation (CV) of the limits (quartiles) among Ecosystem Functional Type (EFT) classes as the number of years included in the study period increases. For each quartile, we plotted the maximum interannual CV (Y axis) among the n consecutive years considered, with n ranging from n=2 to n=8 (X axis). The quartiles of EVI_Mean (our surrogate for productivity) required at least 14 years to stabilize around 5% of CV. The quartiles of EVI_SD (our surrogate for seasonality) required at least 17 years to stabilize around 10% of CV.

**Fig. 3.**

Table 1. Annual quartile boundaries (percentil P25, percentil P50, percentil P75) for EVI_mean and EVI_SD and summary of the period (Interannual mean, Standard Deviation (SD) and Coefficient of Variation (CV)).

| YEAR | EVI_mean P25 | EVI_mean P50 | EVI_mean P75 | EVI_SD P25 | EVI_SD P50 | EVI_SD P75 |
|---|---|---|---|---|---|---|
| 2001 | 0.133 | 0.187 | 0.245 | 0.030 | 0.044 | 0.063 |
| 2002 | 0.139 | 0.190 | 0.243 | 0.031 | 0.042 | 0.057 |
| 2003 | 0.130 | 0.184 | 0.242 | 0.031 | 0.046 | 0.068 |
| 2004 | 0.142 | 0.197 | 0.251 | 0.032 | 0.047 | 0.068 |
| 2005 | 0.123 | 0.168 | 0.222 | 0.023 | 0.039 | 0.056 |
| 2006 | 0.126 | 0.174 | 0.229 | 0.030 | 0.046 | 0.066 |
| 2007 | 0.142 | 0.184 | 0.232 | 0.028 | 0.038 | 0.051 |
| 2008 | 0.133 | 0.176 | 0.229 | 0.029 | 0.042 | 0.062 |
| 2009 | 0.133 | 0.180 | 0.235 | 0.032 | 0.048 | 0.070 |
| 2010 | 0.139 | 0.190 | 0.242 | 0.034 | 0.048 | 0.072 |
| 2011 | 0.149 | 0.200 | 0.258 | 0.032 | 0.045 | 0.069 |
| 2012 | 0.139 | 0.187 | 0.238 | 0.027 | 0.037 | 0.052 |
| 2013 | 0.142 | 0.197 | 0.258 | 0.032 | 0.044 | 0.063 |
| 2014 | 0.130 | 0.184 | 0.241 | 0.026 | 0.037 | 0.056 |
| 2015 | 0.139 | 0.194 | 0.245 | 0.030 | 0.042 | 0.060 |
| 2016 | 0.134 | 0.182 | 0.233 | 0.024 | 0.036 | 0.054 |
| 2017 | 0.142 | 0.187 | 0.238 | 0.030 | 0.039 | 0.057 |
| 2018 | 0.145 | 0.206 | 0.264 | 0.032 | 0.047 | 0.068 |
| Interannual mean | 0.137 | 0.187 | 0.241 | 0.030 | 0.043 | 0.062 |
| Interannual SD | 0.007 | 0.009 | 0.011 | 0.003 | 0.004 | 0.006 |
| Interannual CV (%) | 5.001 | 5.103 | 4.593 | 10.040 | 9.597 | 10.745 |

**Fig. 4.** Table 1